# Improved Training Technique for Shortcut Models

Anh Nguyen*    Viet Nguyen*†    Duc Vu    Trung Dao    Chi Tran    Toan Tran    Anh Tran

Qualcomm AI Research‡

## Abstract

Shortcut models represent a promising, non-adversarial paradigm for generative modeling, uniquely supporting one-step, few-step, and multi-step sampling from a single trained network. However, their widespread adoption has been stymied by critical performance bottlenecks. This paper tackles the five core issues that held shortcut models back: (1) the hidden **flaw of compounding guidance**, which we are the first to formalize, causing severe image artifacts; (2) **inflexible fixed guidance** that restricts inference-time control; (3) a pervasive **frequency bias** driven by a reliance on low-level distances in the direct domain, which biases reconstructions toward low frequencies; (4) **divergent self-consistency** arising from a conflict with EMA training; and (5) **curvy flow trajectories** that impede convergence. To address these challenges, we introduce iSM, a unified training framework that systematically resolves each limitation. Our framework is built on four key improvements: *Intrinsic Guidance* provides explicit, dynamic control over guidance strength, resolving both compounding guidance and inflexibility. A *Multi-Level Wavelet Loss* mitigates frequency bias to restore high-frequency details. *Scaling Optimal Transport (sOT)* reduces training variance and learns straighter, more stable generative paths. Finally, a *Twin EMA* strategy reconciles training stability with self-consistency. Extensive experiments on ImageNet $256 \times 256$ demonstrate that our approach yields substantial FID improvements over baseline shortcut models across one-step, few-step, and multi-step generation, making shortcut models a viable and competitive class of generative models.

## 1 Introduction

Recent generative diffusion models based on flow matching have achieved remarkable success in synthesizing high-fidelity data across various domains [14, 32, 58]. While demonstrably successful, a significant bottleneck persists: generating samples requires integrating the learned vector field over many discrete timesteps, resulting in high computational costs that limit deployment in resource-constrained environments and latency-sensitive applications. Accelerating sampling without sacrificing quality has thus become a critical research area.

Various acceleration techniques have emerged, including timestep distillation [42, 11, 66, 65, 8], advanced numerical solvers [40], lightweight architectures [7, 61, 64], and single-stage training procedures [54, 69, 18]. Shortcut models (SM) [18] offer a particularly elegant approach, using a generator conditioned on both noise level $t$ and desired step size $d$ with an additional self-consistency loss. This enables the prediction of multiple timesteps ahead in a single forward pass, supporting variable sampling budgets at inference using the same network. Despite this promising method, widespread adoption has been hindered by critical performance bottlenecks. This paper tackles the

---

*Equal contribution

†Work done while at Qualcomm.

‡Qualcomm AI Research is an initiative of Qualcomm Technologies, Inc.

39th Conference on Neural Information Processing Systems (NeurIPS 2025).

five core issues that held shortcut models back: *the hidden flaw of compounding guidance, inflexible fixed guidance, frequency bias, divergent self-consistency, and curvy flow trajectories.*

The first two issues stem from a flawed integration of Classifier-Free Guidance (CFG). Not only are users locked into a fixed guidance scale at training time—sacrificing control over the diversity-fidelity trade-off—but this design also conceals a deeper flaw. We are the first to formalize that this fixed guidance *compounds exponentially* across the implicit sub-steps of a large generation step, causing severe image artifacts. The other three issues further degrade sample quality: a reliance on pixel-wise losses creates a *frequency bias* toward blurry images; random noise-data pairings create unstable, *high-curvature generative paths*; and a *temporal lag in the EMA target* prevents the model from learning true self-consistency for large jumps.

To address these challenges, we introduce the **Improved Shortcut Model (iSM)**, a unified training framework that systematically resolves each limitation through four key components.

1. **Intrinsic Guidance**: We resolve both guidance-related flaws by making the guidance scale an explicit input to the model. This provides *dynamic, inference-time control*, works out-of-the-box for one-step generation, and reduces inference time by $\sim 50\%$ compared to standard CFG, all while correcting the compounding effect.

2. **Multi-Level Wavelet Loss**: We replace standard pixel-wise objectives with a *frequency-aware loss* that forces the model to reconstruct high-frequency details, mitigating frequency bias.

3. **Scaling Optimal Transport (sOT)**: We straighten generative trajectories by periodically pooling samples from several mini-batches to compute a large-scale transport plan. This *decouples the OT batch size from the training batch size*, yielding more stable paths without a heavy computational cost.

4. **Twin EMA Strategy**: We eliminate target lag by *maintaining two EMA networks*—a fast-decay one to generate fresh, up-to-date consistency targets and a slow-decay one to ensure stable, high-quality inference.

With these improvements, iSM achieves FID scores of 5.27 and 2.05 on ImageNet $256 \times 256$ with just one and four sampling steps, respectively. Our work *closes the performance gap* with leading generative models and establishes improved shortcut models as a *flexible, efficient, and highly competitive* modeling paradigm.

## 2 Preliminaries

**Flow Matching (FM)** [37, 38] offers an elegant framework for generative modeling of data distributions $p_{\text{data}}(\boldsymbol{x})$. At its core, FM defines $\boldsymbol{x}_t = (1-t)\boldsymbol{x}_0 + t\boldsymbol{x}_1$ as a linear interpolation between a noise sample $\boldsymbol{x}_0$ drawn from a standard normal distribution $\mathcal{N}(0, \mathbf{I})$, denoted $\mathcal{N}$, and a data sample $\boldsymbol{x}_1$ drawn from the data distribution $D$. Here, $t \in [0, 1]$ represents the timestep, parameterizing the interpolation from noise ($t = 0$) to data ($t = 1$). It then trains a velocity model $\bar{\boldsymbol{v}}_\theta(\boldsymbol{x}_t, t, c)$ to match the ground-truth velocity field $\boldsymbol{v} = \boldsymbol{x}_1 - \boldsymbol{x}_0$ by minimizing the following velocity loss:

$$\mathcal{L}_{\text{velocity}}(\theta) = \mathbb{E}_{\substack{\boldsymbol{x}_0 \sim \mathcal{N}, (\boldsymbol{x}_1, c) \sim D \\ t \sim p(t)}} \big[||\bar{\boldsymbol{v}}_\theta(\boldsymbol{x}_t, t, c) - \boldsymbol{v}||^2\big]. \tag{1}$$

Here, $\boldsymbol{x}_t$ is the sample interpolated at time $t$, and $c$ represents any conditioning information, such as text or class labels, associated with the data sample $\boldsymbol{x}_1$. Sampling from such a model typically involves discretizing the learned ODE using numerical methods like Euler integration, often requiring dozens or hundreds of steps. Naively taking large steps with $\bar{\boldsymbol{v}}_\theta$ leads to significant discretization errors, as the predicted velocity points towards an average of potential target data points, causing mode collapse.

**Shortcut Models (SM).** To address this limitation and enable efficient few-step and one-step generation, [18] proposes training a shortcut model that conditions the neural network $\boldsymbol{s}_\theta(\boldsymbol{x}_t, t, c, d)$ not only on the current timestep $t$ and the condition $c$, but also on a desired step size $d$. This allows the model to predict the normalized displacement needed to directly reach the next point $\boldsymbol{x}'_{t+d}$ at time $t + d$, thereby bypassing intermediate steps of the probability flow ODE.

The training approach includes *flow-matching* for infinitesimal steps and *self-consistency* for larger steps. At infinitesimal step sizes ($d \approx 0$), shortcut models use the flow-matching objective, regressing

the model $s_\theta(\boldsymbol{x}_t, t, c, d = 0)$ to predict the empirical velocity $s_{\text{velocity}}$, similar to traditional flow-matching models. For larger step sizes ($d > 0$), shortcut models leverage a self-consistency property where one large shortcut step is equivalent to two consecutive smaller shortcut steps of half the size. This property allows the model to learn efficient large-step transitions by recursively breaking them into smaller steps. To implement this self-consistency objective for training, they select a base number of steps $N$ ($N = 128$ in practice), which defines the smallest time unit for the ODE approximation. This results in $\log_2(N) + 1 = 8$ distinct shortcut lengths available during training, specifically $d \in \{1/128, 1/64, \ldots, 1/2, 1\}$, over which the self-consistency loss is applied. When $d$ is at the smallest value (e.g., 1/128), they instead query the model at $d = 0$. CFG [24], widely used to improve conditional sample quality, is integrated within SM to form guided targets $\boldsymbol{g}_\theta^w$ based on the model output $s_\theta$:

$$\boldsymbol{g}_\theta^w(\boldsymbol{x}_t, t, c, d) = s_\theta(\boldsymbol{x}_t, t, c, d) + w \cdot (s_\theta(\boldsymbol{x}_t, t, c, d) - s_\theta(\boldsymbol{x}_t, t, \varnothing, d)), \quad (2)$$

where $\varnothing$ denotes the null condition. Using the guidance-scaled output $\boldsymbol{g}_\theta^w$, the self-consistency target $s_{\text{consistency}}$ is constructed by simulating two consecutive steps of size $d$:

$$s_{\text{consistency}} = \boldsymbol{g}_\theta^w(\boldsymbol{x}_t, t, c, d)/2 + \boldsymbol{g}_\theta^w(\boldsymbol{x}'_{t+d}, t + d, c, d)/2, \quad (3)$$

where $\boldsymbol{x}'_{t+d} = \boldsymbol{x}_t + \boldsymbol{g}_\theta^w(\boldsymbol{x}_t, t, c, d)d$. These components then form a unified loss function:

$$\mathcal{L}^S(\theta) = \mathbb{E}_{\substack{\boldsymbol{x}_0 \sim \mathcal{N}, (\boldsymbol{x}_1, c) \sim D \\ (t,d) \sim p(t,d)}} \left[ \underbrace{\|s_\theta(\boldsymbol{x}_t, t, c, d = 0) - s_{\text{velocity}}\|^2}_{\text{Flow-Matching}} + \underbrace{\|s_\theta(\boldsymbol{x}_t, t, c, 2d) - s_{\text{consistency}}\|^2}_{\text{Self-Consistency}} \right].$$
$$(4)$$

This approach allows shortcut models to achieve effective multi-step, few-step, and single-step generation using a single trained network.

## 3 Improved Guidance Sampling for Shortcut Models

Shortcut models [18] represent a significant step toward efficient one-step, few-step, and many-step generation. However, their integration of CFG introduces unique challenges compared to standard diffusion methods. This section revisits the shortcut model formulation to identify and address two critical limitations: (1) *inflexibility due to fixed guidance scales*, (2) an *overlooked flaw of accumulation* when applying CFG.

### 3.1 Problem Statement

**Inflexibility of Fixed Guidance.** The original shortcut model framework requires using a fixed guidance scale $w$ during training. This approach is restrictive for two main reasons. First, selecting an optimal $w$ is a difficult hyperparameter tuning problem that depends on the model's final behavior and the intended number of inference steps. Second, and more critically, it removes a key control for balancing the fidelity-diversity trade-off at inference time, substantially reducing the model's *flexibility in production settings*.

**Flaw of Accumulated Guidance.** A critical, previously overlooked issue in shortcut models is the accumulation of guidance effects. We are the first to identify and formalize this critical flaw, which stems from the recursive application of a fixed guidance scale. When a shortcut model generates a sample in a single large step, it implicitly combines the effects of many smaller, guided steps. This *compounding characteristic* leads to an *unintended amplification* of the guidance signal.

To formalize this, we consider a shortcut model $s_\theta$ and its corresponding classifier-free guided output $\boldsymbol{g}_\theta^w$ with guidance scale $w$. To analyze the generation process, we define a sequence of intermediate points $\{\boldsymbol{x}'_{\frac{i}{N}}\}_{i=0}^N$. This sequence is constructed by recursively applying $N$ consecutive shortcut steps, each with the smallest size $d = 1/N$. The process begins from the initial noise $\boldsymbol{x}_0$ at $t = 0$, conditioned on $c$. The starting point of the sequence is thus defined as $\boldsymbol{x}'_0 = \boldsymbol{x}_0$, and subsequent intermediate points are generated as follow:

$$\boldsymbol{x}'_{\frac{i+1}{N}} = \boldsymbol{x}'_{\frac{i}{N}} + s_\theta\left(\boldsymbol{x}'_{\frac{i}{N}}, \frac{i}{N}, c, d\right)d, \quad \text{for } i = 0, \ldots, N - 1. \quad (5)$$

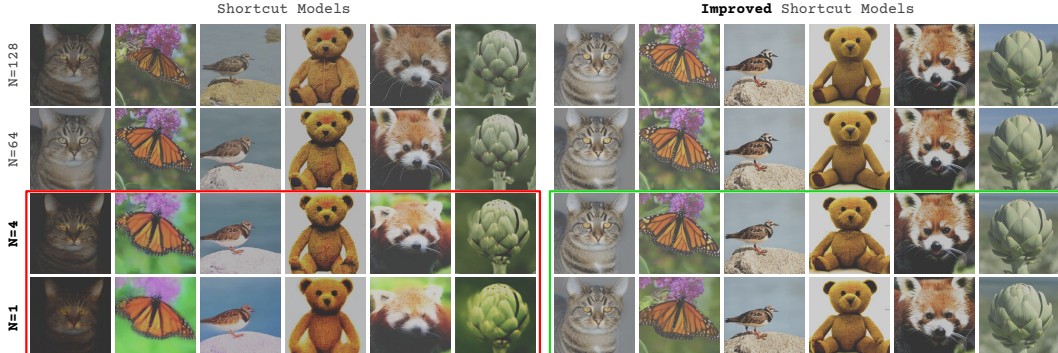

Figure 1: Visualizing the *accumulated guidance* problem in standard shortcut models (left) versus our integrated-guidance approach (right). Top row shows multi-step generation; bottom row shows one-step generation. In the original shortcut model, the fixed guidance strength learned at training time effectively compounds across the many implicit sub-steps, leading to oversaturated colors and blurriness. By conditioning the network on the guidance scale and training it to apply guidance explicitly at each step, our method produces consistent, artifact-free images even in the extreme one-step and few-step regimes.

---

**Proposition 1.** *The model's prediction for a single large shortcut step of size $Nd = 1$ approximately equals the average of the guided displacements corresponding to the $N$ smallest steps, but with an exponentially compounded guidance scale:*

$$\boldsymbol{s}_\theta(\boldsymbol{x}_0, 0, c, Nd) \approx \frac{1}{N} \sum_{i=0}^{N-1} \boldsymbol{g}_\theta^{w^{\log_2(N)}} \left( \boldsymbol{x}'_{\frac{i}{N}}, \frac{i}{N}, c, d \right). \tag{6}$$

*Proof.* See Appendix A. $\qquad\qquad\qquad\qquad\qquad\qquad\qquad\qquad\qquad\qquad\qquad\qquad\quad\square$

---

The original shortcut model [18] defines a fixed CFG scale $w = 1.5$, exclusively applied when constructing self-consistency targets in Eq. (3). Consequently, the one-step generation $\boldsymbol{s}_\theta(\boldsymbol{x}_0, 0, c, 1)$ implicitly aggregates the cumulative effect of $N$ intermediate steps. As formalized in Proposition 1, the guidance scale at each of these implicit intermediate steps is not $w$ but a much higher value $w' = w^{\log_2(N)}$. For $N = 128$ base steps and $w = 1.5$, this calculation yields an extremely high intermediate guidance scale for single large shortcut step cases ($w' = 1.5^{\log_2(128)} \approx 17$). Such high implicit guidance causes artifacts like *over-saturation*, particularly in one-step and few-step generations (see Fig. 1).

### 3.2 Intrinsic Guidance Training for Shortcut Models

To address these limitations of shortcut models [18], we introduce an *Intrinsic Guidance* training framework. In contrast to CFG implementations that operate between model outputs, as commonly practiced in multi-step diffusion models, we integrate the guidance mechanism into the *network's internal space*. The core idea is to make the guidance scale $w$ an *explicit input* to the model and to train the model to produce *guided outputs directly* across a range of scales. This framework enables flexible CFG for few-step and many-step generation and, importantly, provides a principled way to apply CFG for one-step generation while resolving the accumulated guidance issue.

**Model Parameterization.** We define the velocity field generator as a neural network $\boldsymbol{s}_\theta$. Unlike previous shortcut models [18], $\boldsymbol{s}_\theta$ receives the CFG scale $w \geqslant 0$ as an additional input, alongside the current sample $\boldsymbol{x}_t$, time $t$, condition $c$, and the target step size $d$. The network is thus trained to directly output the CFG-modulated velocity $\boldsymbol{s}_\theta(\boldsymbol{x}_t, t, c, d, w)$, allowing guided sample generation with a single network evaluation per step. For implementation details, please refer to Appendix C.

**Flow Matching Objective.** This objective [37, 38] guides the model to predict the conditional and unconditional velocity for infinitesimal steps ($d = 0$) without guidance ($w = 0$), establishing the

base vector fields upon which guidance will be built:

$$\mathcal{L}_{\text{velocity}}(\theta) := \mathbb{E}_{\substack{\boldsymbol{x}_0 \sim \mathcal{N}, (\boldsymbol{x}_1, c) \sim D \\ t \sim p(t)}} \left[ \| \boldsymbol{s}_\theta(\boldsymbol{x}_t, t, c, d = 0, w = 0) - \boldsymbol{s}_{\text{velocity}} \|^2 \right], \tag{7}$$

$$\text{where} \quad \boldsymbol{x}_t = (1 - t)\boldsymbol{x}_0 + t\boldsymbol{x}_1 \quad \text{and} \quad \boldsymbol{s}_{\text{velocity}} = \boldsymbol{x}_1 - \boldsymbol{x}_0.$$

When sampling the condition $c$ for training, we randomly include the null condition $\varnothing$ via stochastic dropout, following the standard scheme [41].

**Intrinsic Guidance Objective.** This objective guides the model to directly produce the CFG-scaled output for infinitesimal steps ($d = 0$) when being conditioned on non-zero guidance strengths ($w > 0$). We begin by recalling the standard formulation of CFG with $w > 0$:

$$\boldsymbol{s}_\theta(\cdot, c, d = 0, w) \approx \boldsymbol{s}_\theta(\cdot, c, d = 0, w = 0) + w \cdot \underbrace{(\boldsymbol{s}_\theta(\cdot, c, d = 0, w = 0) - \boldsymbol{s}_\theta(\cdot, \varnothing, d = 0, w = 0))}_{\boldsymbol{s}_{\text{guidance}}},$$

$$\tag{8}$$

where $\boldsymbol{s}_{\text{guidance}}$ is the estimated guidance direction using the model's base prediction ($w = 0$).

We substitute the unguided velocity $\boldsymbol{s}_\theta(\cdot, c, d = 0, w = 0)$ in Eq. (7) based on the approximation of $\boldsymbol{s}_\theta(\cdot, c, d = 0, w = 0)$ in Eq. (8):

$$\boldsymbol{s}_\theta(\cdot, c, d = 0, w = 0) \approx \boldsymbol{s}_\theta(\cdot, c, d = 0, w) - w \cdot \boldsymbol{s}_{\text{guidance}},$$

$$\| \boldsymbol{s}_\theta(\cdot, c, d = 0, w = 0) - \boldsymbol{s}_{\text{velocity}} \|^2 \approx \| (\boldsymbol{s}_\theta(\cdot, c, d = 0, w) - w \cdot \boldsymbol{s}_{\text{guidance}}) - \boldsymbol{s}_{\text{velocity}} \|^2$$

$$= \| \boldsymbol{s}_\theta(\cdot, c, d = 0, w) - (\boldsymbol{s}_{\text{velocity}} + w \cdot \boldsymbol{s}_{\text{guidance}}) \|^2.$$

Integrating this derived target into Eq. (7), we define a new training objective called the intrinsic guidance loss $\mathcal{L}_{\text{guidance}}$. Crucially, we apply the stop-gradient operator $\text{sg}(\cdot)$ to the guidance direction $\boldsymbol{s}_{\text{guidance}}$ within this loss. This *isolates the optimization* to focus solely on how the network learns to scale its output in response to $w > 0$, preventing interference with the optimization of the foundational $w = 0$ predictions in $\mathcal{L}_{\text{velocity}}$ and improving training stability.

$$\mathcal{L}_{\text{guidance}}(\theta) := \mathbb{E}_{\substack{\boldsymbol{x}_0 \sim \mathcal{N}, (\boldsymbol{x}_1, c) \sim D \\ (t, w) \sim p(t, w)}} \left[ \| \boldsymbol{s}_\theta(\boldsymbol{x}_t, t, c, d = 0, w) - (\boldsymbol{s}_{\text{velocity}} + w \cdot \text{sg}(\boldsymbol{s}_{\text{guidance}})) \|^2 \right], \tag{9}$$

where guidance scale $w > 0$ is drawn from a distribution $p(w)$. This loss trains the network to learn how CFG works for infinitesimal steps ($d = 0$).

**Guided Self-Consistency Objective.** This objective generalizes the self-consistency principle from [18] to operate with arbitrary step sizes ($d > 0$) and any guidance scale ($w \geqslant 0$). The objective maintains the foundational properties of shortcut models, where a *single, large guided shortcut step* yields an output consistent with the composition of *two smaller, consecutive guided steps*.

$$\mathcal{L}_{\text{consistency}}(\theta) := \mathbb{E}_{\substack{\boldsymbol{x}_0 \sim \mathcal{N}, (\boldsymbol{x}_1, c) \sim D \\ (t, w, d) \sim p(t, w, d)}} \left[ \| \boldsymbol{s}_\theta(x_t, t, c, 2d, w) - s_{\text{consistency}} \|^2 \right], \tag{10}$$

$$\text{where} \quad s_{\text{consistency}} := s_{\theta^-}(x_t, t, c, d, w)/2 + s_{\theta^-}(x'_{t+d}, t, c, d, w)/2$$

$$\text{and} \quad x'_{t+d} = x_t + s_\theta(x_t, t, c, d, w)d,$$

where $\theta^-$ is the EMA target network. The stop-gradient operator $\text{sg}(\cdot)$ is applied to the entire consistency target to stabilize training, following standard practice for self-consistency objectives.

**Final Training Objective.** The final objective is a weighted sum of the individual components as follows:

$$\mathcal{L}_{\text{total}}(\theta) = \alpha \mathcal{L}_{\text{velocity}}(\theta) + \beta \mathcal{L}_{\text{guidance}}(\theta) + \gamma \mathcal{L}_{\text{consistency}}(\theta), \tag{11}$$

where $\alpha, \beta, \gamma > 0$ are hyperparameters. For simplicity, we set $\alpha = \beta = \gamma = 1$ in our experiments.

### 3.3 Interval Guidance in Training

Findings presented in [31] suggest that during the inference of diffusion models, strong classifier-free guidance can be *detrimental during the early stages* of the reverse process, corresponding to high noise levels (time $t$ approaching 0 under the convention where $t = 0$ is pure noise).

At high noise levels, the signal-to-noise ratio is low. Hence, the unconditional prediction $s_\theta(x_t, t, \varnothing, d = 0, w = 0)$ points towards the average image in the dataset $D$, while the conditional prediction $s_\theta(x_t, t, c, d = 0, w = 0)$ points vaguely towards the average image for that condition. Consequently, the guidance vector $s_{\text{guidance}}$ points roughly from the global data mean towards the conditional data mean. Applying strong extrapolation ($w > 0$) along this coarse direction to a high-entropy state $x_t$ can cause the sample diversity to *collapse prematurely* towards a few dominant modes.

To mitigate this issue, we incorporate this insight into the training objective $\mathcal{L}_{\text{guidance}}$ in Eq. (9) by applying guidance only within a defined time interval $[t_{\text{interval}}, 1)$. In particular, we modify the guidance scale $w$ used within the Eq. (9) to be time-dependent:

$$w(t) = \begin{cases} w, & \text{if } t \in [t_{\text{interval}}, 1) \\ 0, & \text{otherwise,} \end{cases} \tag{12}$$

where $w$ is the sampled guidance scale. This strategy trains the model to apply zero guidance at high noise levels, preserving diversity in the critical early phase of generation. We find that $t_{\text{interval}} = 0.3$ works well empirically, and this threshold is represented by the red dashed line in Figs. 2b and 2c. Notably, in the high-noise regime where $t < 0.3$ (i.e., before guidance is applied), both flow-matching and self-consistency losses exhibit lower loss pattern.

## 4  Multi-Level Wavelet Function Against Frequency Bias

Deep neural networks often prioritize learning *low-frequency information over high-frequency details*, a phenomenon known as frequency bias [47, 50, 3, 62, 63]. This bias is detrimental to generative models [10, 17, 29, 51], particularly shortcut models [18] optimized for few-step generation with low-level direct domain losses, such as the $\ell_2$ loss. These models typically capture global structure but lack fine-grained, high-frequency details, resulting in blurred textures. The low-level $\ell_2$ loss contributes to this, as it is often dominated by errors in low-frequency components [68].

To mitigate this frequency bias, we introduce optimizing the reconstruction objective in the *wavelet domain*. Specifically, discrete wavelet transform (DWT) is employed to decompose both the prediction and the target (Eqs. (7), (9) and (10)) into their respective wavelet coefficients, yielding multi-band representations, which introduce a *frequency-aware error signal*. By isolating reconstruction errors at different frequency levels, the optimization process is better guided to preserve high-frequency details that are often neglected in standard losses. Consequently, the model learns to generate outputs with significantly improved perceptual quality, capturing both the global structure and fine-grained details.

To further refine supervision across the frequency spectrum, we extend this to a multi-level DWT decomposition. This process involves recursively decomposing the wavelet sub-bands from the previous level. Our experiments consistently show that using a deeper multi-level decomposition, with five levels ($L = 5$) in our experiments, substantially improves the synthesis of high-frequency content compared to a single-level DWT loss or standard pixel-wise objectives. Pseudocode is provided in Appendix B.

## 5  Scaling Optimal Transport for Better Trajectories

Shortcut models, similar to standard flow matching, often encounter considerable loss variance during training. This instability, as highlighted in prior works [46, 33], primarily stems from the *conventional random pairing* of initial noise samples $x_0 \sim \mathcal{N}(0, \mathbf{I})$ with target images ($x_1 \sim D$). Such random coupling frequently results in *intersecting forward trajectories* ($x_t$), where similar noise inputs might be mapped to vastly different data targets. Consequently, the learned generative ODE exhibits *high-curvature reverse paths*, posing a significant challenge for shortcut models. Optimal Transport (OT) within mini-batches can mitigate this by re-assigning noise-image pairs to minimize a transport cost, typically $L_2$ distance, yielding smoother trajectories [46, 33].

Although mini-batch OT helps, we recognize that the efficacy of such mini-batch OT is fundamentally linked to the number of samples considered. Theoretical and empirical findings [13] show that OT better approximates the true transport plan as the number of samples $n$ increases. However, scaling

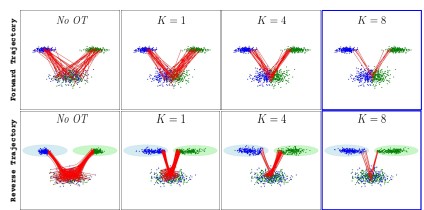
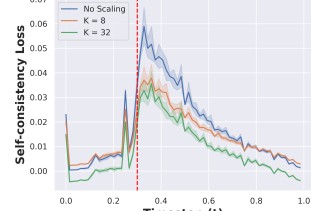
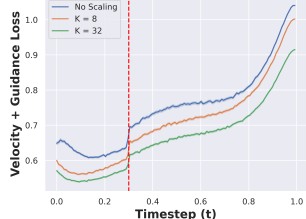

| (a) Forward and reverse trajectories. | (b) Self-consistency Loss. | (c) Velocity & Guidance Losses. |

Figure 2: **Efficacy of Scaling Optimal Transport (sOT) Matching** in improving shortcut model training, demonstrated by varying the OT scaling factor $K$. **(a)** On a bimodal target, forward trajectories (top row) without OT exhibit *frequent intersections* (red), compelling the reverse generative process (bottom row) to follow *high-curvature paths* that initially average the target modes (blue, green). Our sOT approach, by increasing the effective OT scaling $K$, *progressively disentangles these forward couplings*, yielding *substantially straighter reverse trajectories*. **(b, c)** The benefits of this trajectory straightening are reflected in the training losses: both self-consistency (b) and velocity & guidance (c) losses are *consistently lower and more stable with larger $K$*, especially after the interval guidance point $t_{\text{interval}} = 0.3$ (left-side of the red dashed line). **Note for (a)**: To avoid overly dense trajectory, only intersecting forward paths and sampling trajectories with high curvature are highlighted. Zoom for detail.

the mini-batch size $M$ to achieve this is often limited by GPU memory constraints during training. To tackle this scalability challenge, we introduce Scaling Optimal Transport (sOT), a strategy that *decouples the effective batch size for OT computation from the batch size used for model training*. Our method allows us to use a large number of samples for OT without increasing the memory needed for model training. Specifically, for every $K$ training batches of size $M$, we aggregate all noise and image samples into a single set of size $K \times M$ and compute one OT plan over it. The resulting $K \times M$ matched pairs are then split back into $K$ mini-batches of size $M$ and used for the next $K$ standard training steps. This strategy allows us to benefit from an effective OT batch size of $K \times M$ while maintaining a computationally tractable batch size $M$ for model training. The computational overhead from sOT is modest, adding approximately $4\%$ to the total training time in our experiments.

Fig. 2a demonstrates sOT's effectiveness on a 2D bimodal dataset. Random pairing (leftmost column) creates overlapping forward trajectories, forcing the reverse process to follow high-curvature paths that initially move toward an average direction before bending toward target modes. As the effective OT batch size $K$ increases (columns 2-4), sOT produces progressively more *disentangled forward couplings* with fewer intersections, resulting in *straighter generative trajectories*.

Figs. 2b and 2c further illustrate the benefits of sOT by showing a consistent decrease in self-consistency and flow-matching losses, as the sOT scaling factor $K$ increases, with these loss statistics computed over 1.2M ImageNet samples. Regarding self-consistency loss (Fig. 2b), sOT *straightens the learned generative paths* (as seen in Fig. 2a). On these smoother trajectories, a single large shortcut step *more accurately approximates the outcome of two composed smaller steps*, thus reducing the discrepancy and lowering the loss. This indicates that the model learns more reliable long-range predictions. For flow-matching and guidance loss (Fig. 2c), sOT provides more structured noise-data pairings. This results in a less conflicting target velocity field for the model to learn, as similar noise inputs are mapped to more consistently related data targets. This simplified regression task allows the model to learn a more coherent target, thereby reducing the error.

## 6 Twin EMA Improves Self-Consistency Property

Self-consistency objectives seek to align large-step generations with sequences of smaller steps. Specifically, they train an online network $\theta$ to produce outputs from a $2d$-sized step that match the consistency target $s_{\text{consistency}}$, which is obtained by applying two consecutive $d$-sized steps. Instead of using the online network $\theta$ to compute $s_{\text{consistency}}$, the original paper suggests using a target network $\theta^-$, typically implemented as an EMA of $\theta$. When utilized for generating self-consistency targets, EMA parameters provide crucial stability that *dampens the oscillatory behaviors* that would otherwise propagate and amplify across different timesteps.

Despite its essential stabilizing role, conventional EMA implementations introduce a *tension in self-consistency training*. When the target network $\theta^-$ is maintained with a standard slow decay rate, it inevitably represents a historical state of the online network rather than its target distribution. This *temporal lag* means that self-consistency targets $s_{\text{consistency}}$ are derived from *outdated networks*.

Consequently, the online network $\theta$ faces a *conflicting objective*: it must simultaneously optimize for its current trajectory during many-step generation (i.e., flow matching loss) while aligning with targets generated from its historical states during few-step (i.e, self-consistency loss). As training progresses, this misalignment forces the model to learn and balance multiple, potentially contradictory generative mappings rather than converging toward a unified process. The slower the EMA decay rate, the more pronounced this distributional divergence becomes.

To resolve this tension while preserving the essential stabilizing benefits of EMA, we introduce a Twin EMA approach that maintains *two distinct parameter sets*: (1) **Inference Parameters** ($\theta_{\text{infer}}^{-}$): Updated with a conventional slow decay rate and used exclusively during inference, ensuring high-quality sample generation benefits from long-term parameter averaging. (2) **Target Parameters** ($\theta_{\text{target}}^{-}$): Updated with a significantly faster decay rate to maintain close proximity to the online network's current state while still providing stabilization. This *decoupling addresses the fundamental conflict* between stability and recency requirements. By generating self-consistency targets from a near-contemporary version of itself, the online network more effectively enforces consistency across different timestep discretizations. Meanwhile, the separate maintenance of slow-decay parameters for inference preserves the stability crucial for high-quality sample generation.

Table 1: **Quantitative results** on class-conditional ImageNet $256 \times 256$ across different model types.

| Model | FID-50k ($\downarrow$) | NFE ($\downarrow$) | #Params |
|---|---|---|---|
| **GAN** | | | |
| BigGAN-deep [5] | 4.06 | 1 | 112M |
| GigaGAN [26] | 3.45 | 1 | 569M |
| StyleGAN-XL [28] | 2.30 | 1 | 166M |
| **Masked & AR** | | | |
| VQGAN [16] | 26.52 | 1024 | 227M |
| MaskGIT [6] | 6.18 | 8 | 227M |
| VAR-$d$20 [56] | 2.57 | 10 | 600M |
| VAR-$d$30 [56] | 1.92 | 10 | 2B |
| MAR [35] | 1.98 | 100 | 400M |
| **Diffusion & Flow Models** | | | |
| ADM [12] | 10.94 | 250 | 554M |
| CDM [23] | 4.88 | 8100 | - |
| SimDiff [25] | 2.77 | 512 | 2B |
| LDM-4-G [48] | 3.60 | 250 | 400M |
| U-DiT-L [57] | 3.37 | 250 | 916M |
| U-ViT-H [2] | 2.29 | 50 | 501M |
| DiT-XL/2 [44] | 2.27 | 250 | 675M |
| SiT-XL/2 [41] | 2.15 | 250 | 675M |
| REPA-XL/2 [67] | 1.42 | 250 | 675M |
| FlowDCN [60] | 2.00 | 250 | 618M |
| **One-to-Many Step Models** | | | |
| iCT [54] | 34.24 | 1 | 675M |
| | 20.3 | 2 | 675M |
| SM [18] | 10.60 | 1 | 675M |
| | 7.80 | 4 | 675M |
| | 3.80 | 128 | 675M |
| IMM [69] | 7.77 | 1 | 675M |
| | 3.99 | 2 | 675M |
| | 2.51 | 4 | 675M |
| | 1.99 | 8 | 675M |
| **iSM (ours)** | 5.27 | 1 | 675M |
| | 2.44 | 2 | 675M |
| | 2.05 | 4 | 675M |
| | 1.93 | 8 | 675M |
| | 1.88 | 128 | 675M |

## 7 Experiment

We combine all the improvements detailed from Sections 3 to 6 to train our improved shortcut models. We extensively benchmark our models on ImageNet $256 \times 256$ against various models across multiple training paradigms. We evaluate sample quality using FID-50K [20]. For all experiments, our models use the XL/2 variant of the SiT architecture [41]. We report evaluations against other models at *800K iterations*, while component-wise analyses are conducted at *250K iterations*. We summarize our main quantitative benchmarks in Table 1, provide a broader evaluation using additional metrics in Table 2, present a detailed component-wise analysis in Table 3, and demonstrate the scalability and generality of our framework in Table 4.

Table 2: **Broader evaluation** with FD-DINOv2 and IS on ImageNet $256 \times 256$.

| Model | NFE | FD-DINOv2$\downarrow$ | IS$\uparrow$ |
|---|---|---|---|
| SM [18] | 1 | 500.92 | 102.66 |
| IMM [69] | 1 | 247.78 | 128.87 |
| **iSM (ours)** | 1 | **232.31** | **223.52** |
| SM [18] | 2 | 329.53 | 125.66 |
| IMM [69] | 2 | 152.08 | 173.66 |
| **iSM (ours)** | 2 | **107.63** | **302.29** |
| SM [18] | 4 | 265.90 | 136.79 |
| IMM [69] | 4 | 110.88 | 204.95 |
| **iSM (ours)** | 4 | **83.70** | **298.23** |

### 7.1 Quantitative result and comparison

Our model, iSM, achieves strong FID scores across various NFE settings, demonstrating high sample quality and inference efficiency. With 8 inference steps, iSM *surpasses the 10-step VAR* [56] of similar model size and performs *comparably to VAR's much larger 2B variant*. A key advantage of iSM is its inherent support for *flexible NFE counts*—from single-step to many-step generation—a feature absent in models like VAR, with FID scores significantly improving as NFE increases from 1 to 8. Furthermore, iSM performs favorably against other recent variable-step models, including the original shortcut model [18] and IMM [69]. For instance, in single-step generation, iSM attains an

Table 3: **Ablation study** on ImageNet $256 \times 256$. We investigate the impact of key hyperparameters for each of our proposed components. The best-performing setting from each block is carried forward to the next. All models are trained for 250K iterations.

| Method | $w_{\max}$ | $t_{\text{interval}}$ | $L$ | $K$ | $\theta_{\text{target}}^{-}$ | $\text{FID}_{N=1} \downarrow$ | $\text{FID}_{N=4} \downarrow$ |
|---|---|---|---|---|---|---|---|
| Intrinsic Guidance (Sec. 3) | 2.0 | 0 | 0 | 0 | 0.9999 | 10.10 | 3.21 |
| | **3.5** | 0 | 0 | 0 | 0.9999 | 9.62 | 3.17 |
| | 5.0 | 0 | 0 | 0 | 0.9999 | 10.38 | 3.34 |
| Interval Guidance (Sec. 3) | 3.5 | 0.0 | 0 | 0 | 0.9999 | 9.62 | 3.17 |
| | 3.5 | 0.1 | 0 | 0 | 0.9999 | 8.58 | 3.14 |
| | 3.5 | **0.3** | 0 | 0 | 0.9999 | 8.49 | 2.81 |
| | 3.5 | 0.5 | 0 | 0 | 0.9999 | 19.22 | 2.84 |
| Multi-level Wavelet (Sec. 4) | 3.5 | 0.3 | 0 | 0 | 0.9999 | 8.49 | 2.81 |
| | 3.5 | 0.3 | 1 | 0 | 0.9999 | 8.21 | 2.79 |
| | 3.5 | 0.3 | 3 | 0 | 0.9999 | 8.17 | 2.75 |
| | 3.5 | 0.3 | **5** | 0 | 0.9999 | 8.12 | 2.64 |
| Scaling OT (Sec. 5) | 3.5 | 0.3 | 5 | 0 | 0.9999 | 8.12 | 2.64 |
| | 3.5 | 0.3 | 5 | 1 | 0.9999 | 8.07 | 2.51 |
| | 3.5 | 0.3 | 5 | 8 | 0.9999 | 8.03 | 2.28 |
| | 3.5 | 0.3 | 5 | **32** | 0.9999 | 7.97 | 2.23 |
| Twin EMA (Sec. 6) | 3.5 | 0.3 | 5 | 32 | 0.9999 | 7.97 | 2.23 |
| | 3.5 | 0.3 | 5 | 32 | 0.999 | 7.43 | 2.21 |
| | 3.5 | 0.3 | 5 | 32 | **0.95** | 6.56 | 2.16 |

FID of 5.27, *outperforming both*. This strong performance extends to few-step scenarios, with FIDs of 2.05 (4 steps) and 1.93 (8 steps), highlighting iSM's robustness across different inference budgets.

To provide a more comprehensive validation, we extend our evaluation to include FD-DINOv2 and Inception Score (IS). As presented in Table 2, our iSM framework consistently outperforms the baselines across these metrics. The strong FD-DINOv2 scores confirm that our model's high fidelity is robust and not specific to the Inception feature space; for example, at 4 NFE, iSM achieves a score of 83.70, a *greater than 3 times improvement* over the baseline SM's 265.90. The Inception Scores show a particularly large margin of improvement; for example, at 2 inference steps, our model's IS of 302.29 more than *doubles* the baseline score of 125.66. The consistent gains across these distinct evaluation metrics provide strong evidence that iSM yields significant and robust improvements to shortcut model generation.

## 7.2 Component-wise Analysis

We explore whether our proposed techniques enhance shortcut model training. As shown in Table 3, each component contributes to improved generation quality, with their combination achieving a significantly better FID score compared to the vanilla model. Below, we present a detailed analysis of the individual impact of each component.

**Intrinsic Guidance.** We examine the effect of varying the range of the conditioning CFG scale $w$ on model performance. During training, we sample discrete values of $w$ from the interval $[0, w_{\max}]$ using a step size of 0.25. Through empirical evaluation, we find that setting $w_{\max} = 3.5$ leads to the best results. When $w_{\max}$ is set too high (e.g., 5.0), the model is exposed to a broader range of conditioning strengths. This can introduce unnecessary complexity, as the model must learn to generalize over a wider set of conditioning values, potentially degrading overall performance. On the other hand, if $w_{\max}$ is too low (e.g., 2.0), the model only learns from outputs corresponding to low CFG scales, which may lack high quality.

**Interval Guidance in Training.** We examined the effect of varying $t_{\text{interval}}$, the threshold below which CFG is disabled. We observed that increasing $t_{\text{interval}}$ from very low values up to a moderate range (e.g., 0.1 to 0.3) improved both metrics, consistent with the benefit of avoiding strong, coarse guidance in high-noise regimes. However, further increasing $t_{\text{interval}}$ to higher values (e.g., 0.5) results in a significant increase in FID. Training the model without guidance for a substantial portion of the early reverse process may impair its ability to effectively leverage guidance when $t \geqslant t_{\text{interval}}$.

**Multi-level Wavelet Function.** We also investigate the impact of varying the number of decomposition levels in the wavelet transform. Given that the latent representation is of size $32 \times 32$, the maximum feasible number of levels is $\log_2(32) = 5$. Our experiments show that utilizing the full 5

levels yields better performance compared to using fewer levels (e.g., 1 or 3) or not using the wavelet transform entirely.

**Scaling Optimal Transport.** We also examine the impact of OT matching on the performance of our previous improvements. Our results show that using traditional OT matching results in lower FID scores than not using OT. We then assess the effectiveness of our proposed enhanced OT by applying it at two different scales: $K = 8$ and $K = 32$. Our results reveal a clear correlation between increasing OT scale and improved FID scores. Although scaling beyond $K = 32$ may further enhance performance, we select $K = 32$ as a trade-off between computational overhead and effectiveness.

**Twin EMA.** Finally, our proposed Twin EMA was evaluated using decay rates of $0.999$ and $0.95$ for the target network $\theta_{\text{target}}^{-}$. A decay rate of $0.95$ provided an effective balance between training stability and mitigating distributional divergence from self-consistency targets. This was evidenced by a substantial reduction in FID, achieving 6.56 and 2.16 for one-step and four-step generation, respectively.

### 7.3 Generalizing to Higher Resolutions and Architectures

To demonstrate the *architectural generality* and *resolution scalability* of our framework, we applied our **iSM** training to FlowDCN [59], a fully convolutional generative model with group-wise deformable convolution blocks, on ImageNet $512 \times 512$. As shown in Table 4, at 300K training iterations, our method yields a substantial FID improvement over the baseline, confirming its effectiveness across different model families and at higher scales. For instance, in 4-step generation, our approach reduces the FID from 12.16 to 9.94. The improvement is also pronounced in the one-step generation, where iSM improves sample diversity by *boosting Recall over 5 times* from 0.11 to 0.55.

Table 4: **Scalability and generality** of iSM on ImageNet $512 \times 512$ with the FlowDCN architecture, reported after 300K training iterations.

| Model | NFE | FID↓ | Precision↑ | Recall↑ |
|---|---|---|---|---|
| SM | 1 | 43.81 | 0.56 | 0.11 |
| **iSM (ours)** | 1 | **37.05** | **0.60** | **0.55** |
| SM | 4 | 12.16 | **0.86** | 0.19 |
| **iSM (ours)** | 4 | **9.94** | 0.78 | **0.62** |

## 8 Conclusion

This work confronts the foundational obstacles that have previously limited the performance of shortcut models. We systematically address five core challenges: the damaging effects of compounding guidance, inflexible fixed guidance, a pervasive low-frequency bias, conflicts between self-consistency and EMA updates, and instability from high-curvature flow trajectories. To resolve these issues, we present **iSM**, a unified training framework composed of four key components. By incorporating *Intrinsic Guidance* for dynamic control while resolving the flaw of compounding guidance, a *Multi-Level Wavelet Loss* for high-frequency fidelity, *Scaling Optimal Transport (sOT)* for smoother trajectories, and a *Twin EMA* strategy for stable consistency, iSM elevates shortcut models into a competitive paradigm for generative modeling.

Our extensive experiments on ImageNet $256 \times 256$ demonstrate that iSM yields substantial FID improvements over baseline shortcut models across one-step, few-step, and multi-step generation. These results close the performance gap with leading GANs and other diffusion-based models, establishing shortcut models as a viable and competitive alternative that uniquely offers variable sampling budgets from a single network.

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

# A Proof of Proposition 1

Given a shortcut model $s_\theta$ and its corresponding classifier-free guided output $g_\theta^w$ with guidance scale $w$. To analyze the generation process, we define a sequence of intermediate points $\{x'_{\frac{i}{N}}\}_{i=0}^N$. This sequence is constructed by recursively applying $N$ consecutive shortcut steps, each with the smallest step size $d = 1/N$. Note that $N$ is chosen to be a power of 2. The process begins from the initial noise $x_0$ at $t = 0$, conditioned on $c$. The starting point of the sequence is thus defined as $x'_0 = x_0$, and subsequent intermediate points are generated as follow:

$$x'_{\frac{i+1}{N}} = x'_{\frac{i}{N}} + s_\theta\left(x'_{\frac{i}{N}}, \frac{i}{N}, c, d\right) d, \quad \text{for } i = 0, \ldots, N-1. \tag{13}$$

Ideally, we assume that the model is perfectly trained, or equivalently, that the loss in Eq. (4) is minimized to zero. Under this assumption, we have:

$$s_\theta(x'_t, t, c, 2d) = \frac{1}{2}[g_\theta^w(x'_t, t, c, d) + g_\theta^w(x'_{t+d}, t+d, c, d)], \tag{14}$$

$$s_\theta(x'_t, t, \varnothing, 2d) = \frac{1}{2}[s_\theta(x'_t, t, \varnothing, d) + s_\theta(x'_{t+d}, t+d, \varnothing, d)]. \tag{15}$$

In the following derivation, we omit the time $t$ in the network notation for simplicity, unless otherwise specified. First, we will prove by induction that:

$$g_\theta^w(x'_t, c, 2^j d) = \frac{1}{2^j} \sum_{i=0}^{2^j-1} g_\theta^{w^{j+1}}\left(x'_{t+id}, c, d\right). \tag{16}$$

For the base case $j = 0$, we have $g_\theta^w(x'_t, c, d) = g_\theta^w(x'_t, c, d)$, which satisfies Eq. (16). Assuming that Eq. (16) holds for $j = k$, we now show that it also holds for $j = k+1$. Using Eq. (14) and Eq. (15), we obtain:

$$s_\theta(x'_t, c, 2^{k+1}d) = \frac{1}{2}[g_\theta^w(x'_t, c, 2^k d) + g_\theta^w(x'_{t+2^k d}, c, 2^k d)]$$

$$= \frac{1}{2^{k+1}}\left[\sum_{i=0}^{2^k-1} g_\theta^{w^{k+1}}\left(x'_{t+id}, c, d\right) + \sum_{i=0}^{2^k-1} g_\theta^{w^{k+1}}\left(x'_{t+2^k d+id}, c, d\right)\right]$$

$$= \frac{1}{2^{k+1}} \sum_{i=0}^{2^{k+1}-1} g_\theta^{w^{k+1}}\left(x'_{t+id}, c, d\right),$$

$$s_\theta(x'_t, \varnothing, 2^{k+1}d) = \frac{1}{2}\left[s_\theta(x'_t, \varnothing, 2^k d) + s_\theta(x'_{t+2^k d}, \varnothing, 2^k d)\right]$$

$$= \frac{1}{2^{k+1}} \sum_{i=0}^{2^{k+1}-1} s_\theta(x'_{t+id}, \varnothing, d).$$

Therefore, we have:

$$g_\theta^w(x'_t, c, 2^{k+1}d)$$

$$= w s_\theta(x'_t, c, 2^{k+1}d) + (1-w)s_\theta(x'_t, \varnothing, 2^{k+1}d)$$

$$= \frac{1}{2^{k+1}} \sum_{i=0}^{2^{k+1}-1} [w g_\theta^{w^{k+1}}\left(x'_{t+id}, c, d\right) + (1-w)s_\theta(x'_{t+id}, \varnothing, d)]$$

$$= \frac{1}{2^{k+1}} \sum_{i=0}^{2^{k+1}-1} [w(w^{k+1}s_\theta(x'_{t+id}, c, d) + (1-w^{k+1})s_\theta(x'_{t+id}, \varnothing, d)) + (1-w)s_\theta(x'_{t+id}, \varnothing, d)]$$

$$= \frac{1}{2^{k+1}} \sum_{i=0}^{2^{k+1}-1} [w^{k+2}s_\theta(x'_{t+id}, c, d) + (1-w^{k+2})s_\theta(x'_{t+id}, \varnothing, d)]$$

$$= \frac{1}{2^{k+1}} \sum_{i=0}^{2^{k+1}-1} g_\theta^{w^{k+2}}\left(x'_{t+id}, c, d\right).$$

This satisfies Eq. (16) for $j = k + 1$. By induction, we have Eq. (16) holds for all $j$. Based on this result, we can rewrite the output of the shortcut model for a single large step of size $Nd = 1$, denoted by $s_\theta(x_0, c, Nd)$, as follows:

$$s_\theta(x_0, c, Nd) = \frac{1}{2}\left[g_\theta^w\left(x_0, c, \frac{N}{2}d\right) + g_\theta^w\left(x'_{\frac{1}{2}}, c, \frac{N}{2}d\right)\right]$$

$$= \frac{1}{N}\left[\sum_{i=0}^{N/2-1} g_\theta^{w^{\log 2(N/2)+1}}\left(x'_{\frac{i}{N}}, c, d\right) + \sum_{i=0}^{N/2-1} g_\theta^{w^{\log 2(N/2)+1}}\left(x'_{\frac{1}{2}+\frac{i}{N}}, c, d\right)\right]$$

$$= \frac{1}{N}\sum_{i=0}^{N-1} g_\theta^{w^{\log 2(N)}}\left(x'_{\frac{i}{N}}, c, d\right),$$

which completes the proof.

## B   Multi-Level Wavelet Function

**Multi-Level Wavelet Function.** Algorithm 1 provides pseudo-code for our proposed multi-level wavelet objective described in Section 4.

---
**Algorithm 1** Multi-level Wavelet Function
---

```
class MultiLevelWaveletLoss:
    def __init__(self):
        self.dwt = DWT_2D("haar") # Discrete Wavelet Transform (DWT)
        self.diff_func = MSELoss() # Distance function

    def concatenated_dwt(self, x):
        xll, xlh, xhl, xhh = self.dwt(x) # Decompose into 4 wavelet sub-bands using DWT
        details = torch.cat([xll, xlh, xhl, xhh], dim=1) # Concatenate the sub-bands
        return details

    def __call__(self, pred, target, num_levels):
        total_loss = diff_func(pred, target).mean() # Calculate low-level loss in original output space

        # Recursively calculate loss on wavelet sub-bands for each level
        pred_curr, target_curr = pred, target
        for current_level in range(num_levels):
            # Derive predicted and target sub-bands using outputs from previous level
            pred_bands = self.concatenated_dwt(pred_curr)
            target_bands = self.concatenated_dwt(target_curr)

            # Calculate loss on current level
            total_loss += diff_func(pred_bands, target_bands).mean()
            pred_curr, target_curr = pred_bands, target_bands

        # Taking average from all levels
        loss = total_loss / (num_levels + 1)
        return loss
```

---

**Results.** Fig. 3 presents qualitative comparisons illustrating the impact of different levels in our multi-level wavelet function versus traditional low-level loss. Incorporating more wavelet levels yields finer details and fewer artifacts, especially in one- and few-step generation.

## C   Injecting Conditional Inputs into Network

We explore two primary strategies for incorporating conditional information including CFG scale $w$, current sample $x_t$, time $t$, condition $c$, and the target step size $d$ into our network. The first strategy, similar to U-ViT [2], encodes each condition as an individual token appended to the input sequence of noisy image patch tokens. The second strategy employs AdaLN-Zero blocks [44] to modulate the network with each condition; these modulations are then aggregated through addition. We find that the AdaLN-Zero approach yields comparable performance to the U-ViT method but without the drawback of increased input sequence length. Given this efficiency benefit, we adopt AdaLN-Zero for injecting CFG information into our network.

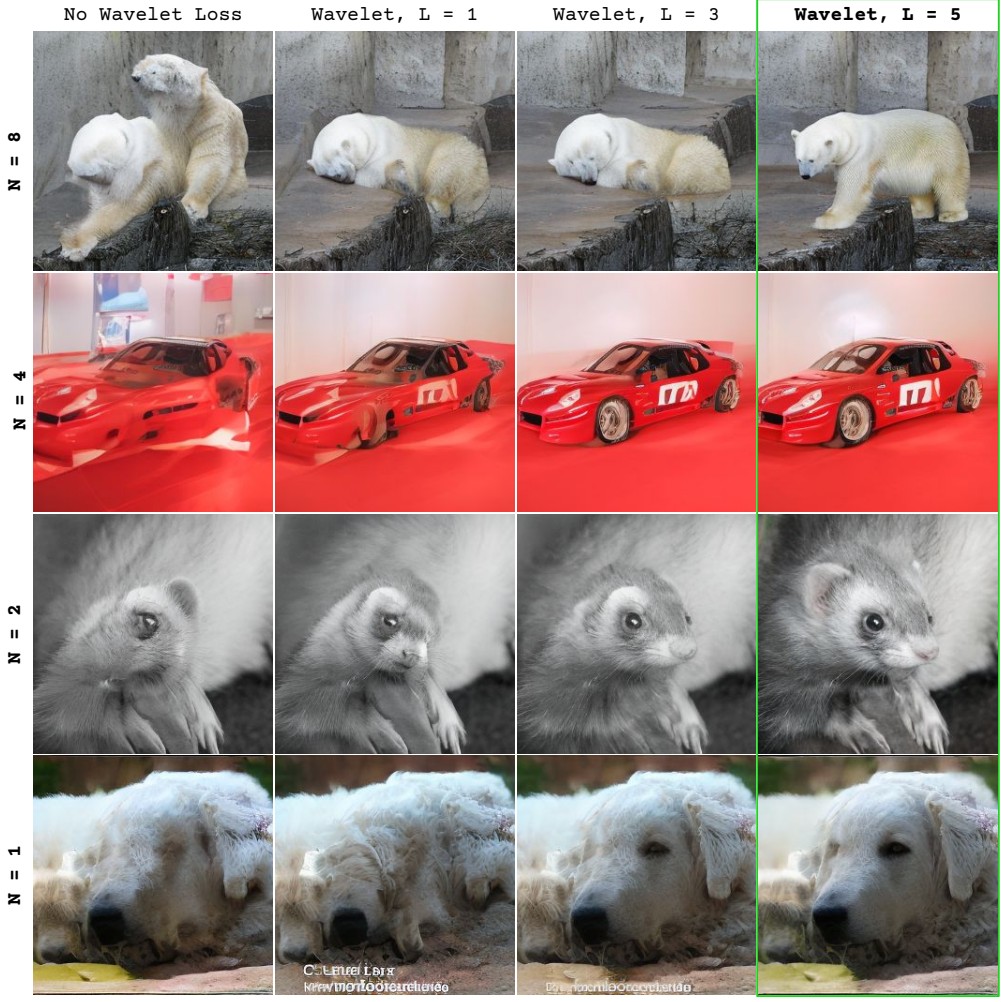

Figure 3: Effect of applying different numbers of wavelet loss layers ($L$) on image quality under one and few-step inference. The traditional low-level objective (leftmost column) results in noticeable degradation and artifacts. In contrast, using multi-level wavelet loss, especially with $L = 5$ (green box, right), produces high-quality, consistent images across all inference settings.

## D  Accelerating training speed

To mitigate the high computational cost of short-cut models, we explore resuming their training from existing a flow matching weights [67]. Table 5 shows that while resuming offers limited improvement for the original shortcut model, our improved shortcut models (iSM) demonstrate considerable gains. Specifically, when incorporating a series of proposed methods, iSM achieves significantly better FID scores. Models are trained for 250k steps for all evaluations.

Table 5: Performance of Improved Shortcut Models

| Method | $\text{FID}_{N=1} \downarrow$ | $\text{FID}_{N=4} \downarrow$ |
|---|---|---|
| Shortcut Models [18] | 21.38 | 13.46 |
| *Improved Shortcut Models (iSM)* | | |
| + Intrinsic Guidance | 9.62 | 3.17 |
| + Interval Guidance in Training | 8.49 | 2.81 |
| + Multi-level Wavelet Function | 8.12 | 2.64 |
| + Scaling Optimal Transport | 7.97 | 2.23 |
| + Twin EMA | 6.56 | 2.16 |

## E  Experiment Settings

### E.1  Training & Parameterization Setting

For experiments on ImageNet, images are preprocessed to 256x256 resolution, following the protocol of ADM [12]. Our models operate in the latent space derived from the sd-vae-ft-mse autoencoder [1]. These latents are subsequently normalized channel-wise using means of $[0.86488, -0.27787343, 0.21616915, 0.3738409]$ and standard deviations of $[4.85503674, 5.31922414, 3.93725398, 3.9870003]$. Finally, the normalized latents are scaled

Table 6: Experimental settings on ImageNet $256 \times 256$.

| **Parameterization Setting** | |
| --- | --- |
| Architecture | SiT-XL |
| GFlops | 118.64 |
| Params (M) | 675 |
| Flow Trajectory | OT-FM |
| Input dim. | $32 \times 32 \times 4$ |
| Num. layers | 24 |
| Hidden dim. | 1024 |
| Num. heads | 16 |
| $\alpha_t$ | $1 - t$ |
| $\sigma_t$ | $t$ |
| $w_t$ | $\sigma_t$ |
| Training objective | v-prediction |
| **Training Setting** | |
| Training iteration | 800K |
| Dropout | 0 |
| Optimizer | AdamW |
| AdamW $\beta_1$ | 0.9 |
| AdamW $\beta_2$ | 0.999 |
| AdamW $\epsilon$ | $10^{-8}$ |
| Learning Rate | 0.0001 |
| Weight Decay | 0 |
| Batch Size | 256 |
| Label Dropout | 0.1 |
| **Methods** | |
| CFG Scale $w_{\max}$ | 3.5 |
| Interval $t_{\text{interval}}$ | 0.3 |
| Wavelet Levels $L$ | 5 |
| OT Scale $K$ | 32 |
| Ratio of Empirical to Self-consistency Targets | 0.25 |
| EMA Parameters Used For Self-consistency Targets? | True |
| EMA Target Rate $\theta_{\text{target}}^-$ | 0.95 |
| EMA Parameters Used For Evaluation? | True |
| EMA Inference Rate $\theta_{\text{infer}}^-$ | 0.9999 |

by a factor of 0.5, targeting an approximate standard deviation of 0.5 for the network input. Detailed configurations for all experiments are provided in Table 6.

## E.2 Evaluation Details

We follow the ADM [12] evaluation setup, using the same reference batches from their official implementation,[4] and compute FID [20] over 50K generated images.

## E.3 Baselines

Below, we summarize the baseline methods used in our evaluation.

- **ADM** [12] improves U-Net-based diffusion architectures and introduces classifier-guided sampling to balance sample quality and diversity.
- **CDM** [23] proposes cascaded diffusion models, which generate images in a coarse-to-fine manner, similar to ProgressiveGAN [27].

---

[4] https://github.com/openai/guided-diffusion/tree/main/evaluations

- **Simple diffusion** [25] leverages a diffusion model for high-resolution images by simplifying the noise schedule and model architectures.
- **LDM** [48] introduces the concept of operating diffusion models in a compressed latent space, enhancing efficiency.
- **U-DiT** [57] proposes a series of U-shaped DiTs based on self-attention with downsampled tokens.
- **U-ViT** [2] adapts Vision Transformers for latent diffusion by incorporating U-Net-like long skip connections.
- **DiT** [44] pioneers the use of a pure transformer architecture as the backbone for diffusion models, featuring AdaIN-zero modules.
- **SiT** [41] investigates how to improve DiT training by transitioning from discrete diffusion to continuous flow-based modeling.
- **REPA** [67] accelerates the training of DiT/SiT models by regularizing network representations to align with features from pretrained visual encoders.
- **FlowDCN** [60] offers a fully convolutional architecture for generative modeling with linear time and memory complexity, enabling efficient high-resolution image synthesis.
- **iCT** [53] introduces several techniques to enhance the training of CMs [54], including a log-normal noise schedule, Pseudo-Huber loss functions, and a scheduler for total discretization steps during training.
- **SM** [18] establishes a framework for one-step generation by combining flow matching with a self-consistency objective.
- **IMM** [69] few-step generative models by inductively matching all moments of bootstrapped samples derived from stochastic interpolants using Maximum Mean Discrepancy (MMD), aiming for stable convergence.

# F    Related Works

## F.1    Diffusion, Consistency & Flow Matching models

Generative modeling has seen significant advancements with methods like diffusion models, consistency models, and flow matching. Diffusion models [52, 55, 22, 30] are a generative framework that synthesizes data by gradually transforming random noise into outputs via a stochastic denoising process. To introduce efficient generation, consistency models [54, 53, 39] aim for efficient generation by learning a mapping from any point on a solution trajectory directly to the data origin. Meanwhile, flow matching [37, 38] proposes learning generative models by defining a target vector field between noise and data and training a neural network to approximate this field. Both these methods have demonstrated strong performance when scaled to text-to-image [48, 49, 45, 9, 15] and text-to-video [21, 4, 43] applications.

## F.2    End-to-end training for one-to-many step generative models

The quest for efficient, high-quality few-step generation has explored several avenues. While early one-step models leveraged GANs [19, 28, 5] and MMD [36, 34], the inherent instability and complexity of adversarial training limit their scalability. Consistency Models (CMs) [54, 53, 39] address the instability of adversarial training without needing synthetic datasets. Unlike distillation, CMs can be trained from scratch via consistency training (CT), independent of pre-trained diffusion models. However, few-step generation with CMs usually involves discrete-time variants. These variants require meticulous timestep scheduling and are susceptible to irreducible bias accumulation due to the inherent ambiguity of discretization. Inductive Moment Matching (IMM) [69] further advanced stable few-step training using moment matching. Shortcut Models (SMs) [18] introduced step-size conditioning with self-bootstrapping. However, original SMs (Section 3) suffer from inflexible guidance and guidance accumulation artifacts, which our work directly addresses.

# G    More Qualitative Results

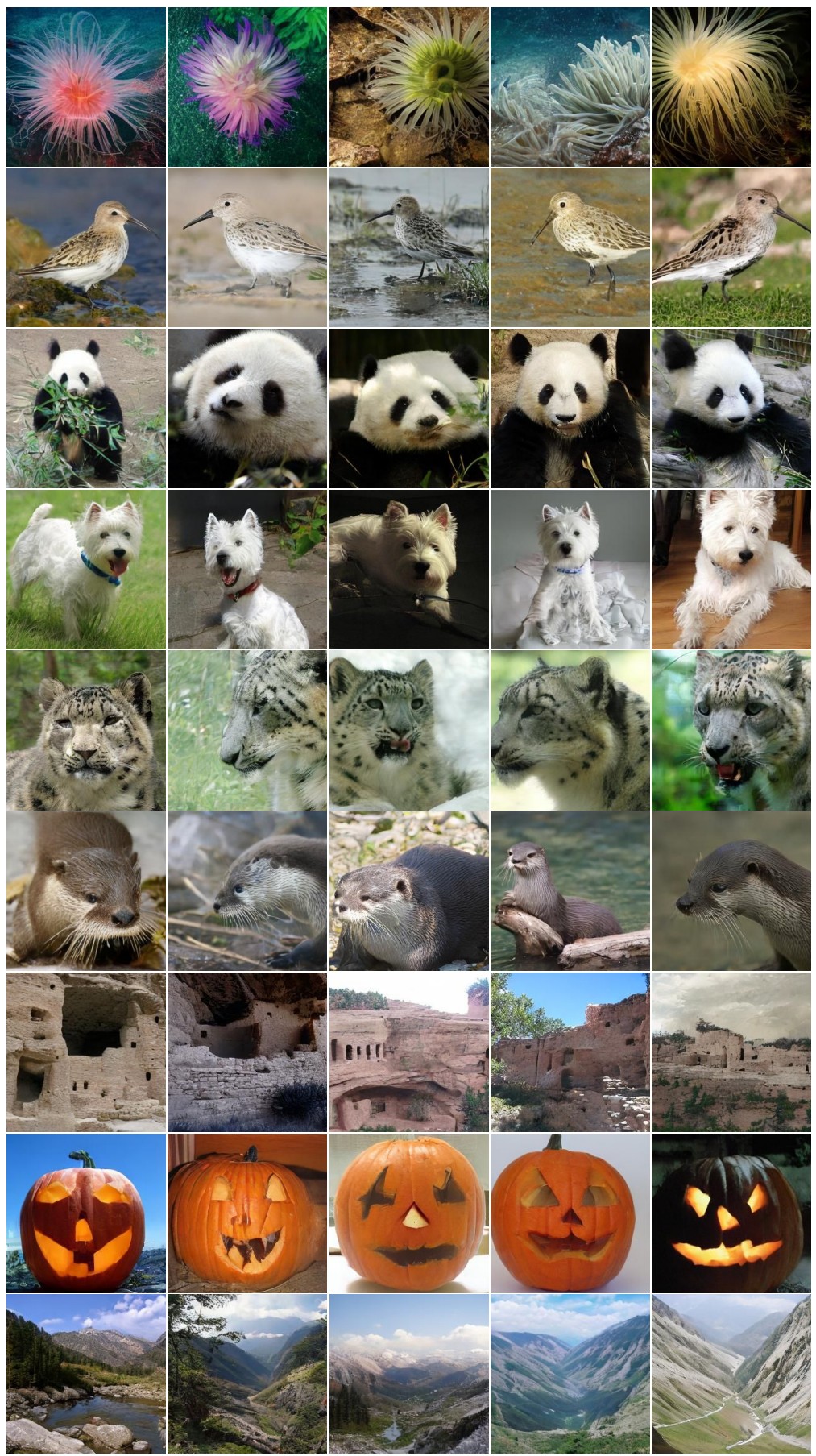

Figure 4: Uncurated samples on ImageNet-256 × 256 with 1-step sampling.

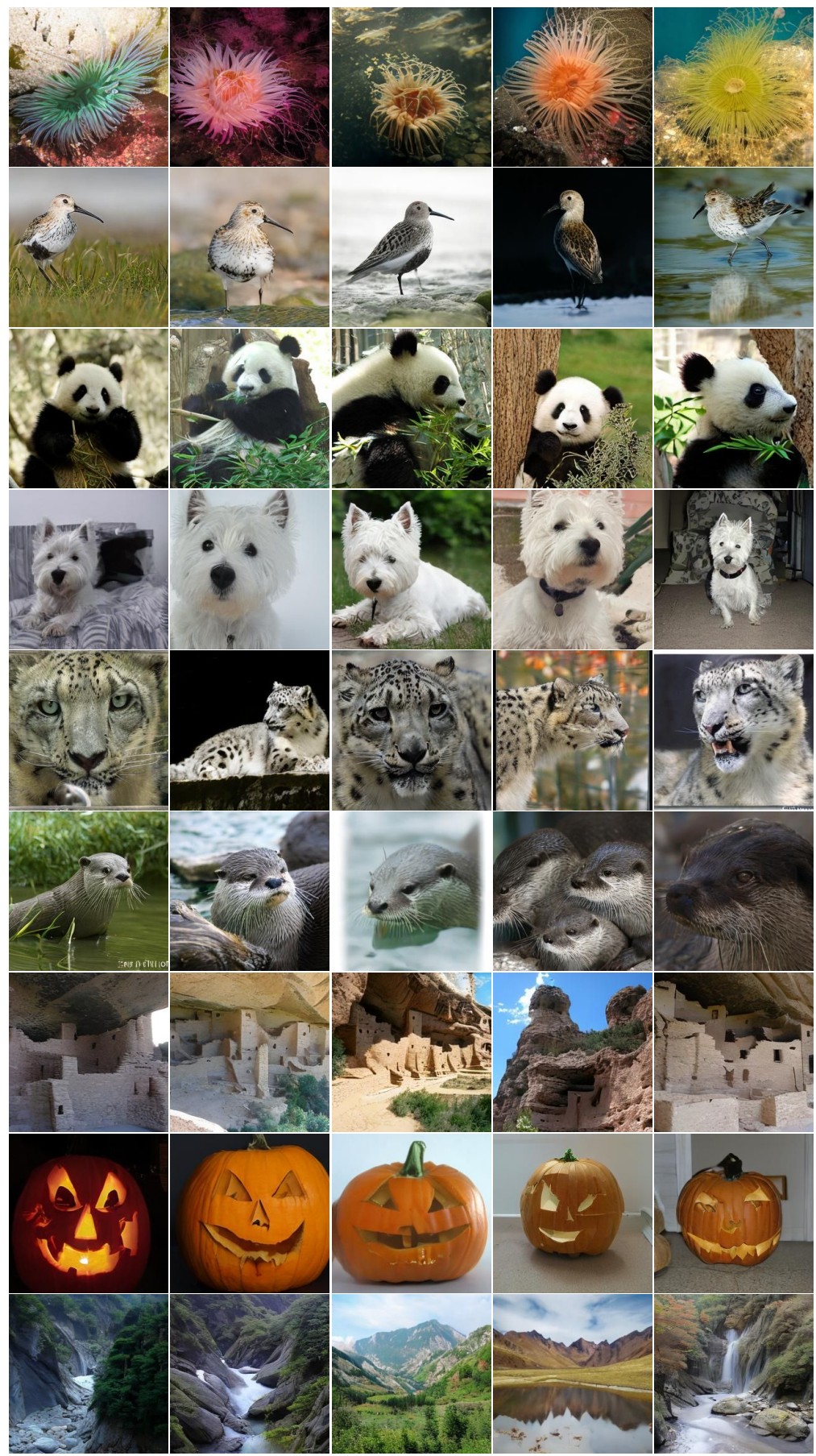

Figure 5: Uncurated samples on ImageNet-256 × 256 with 2-step sampling.

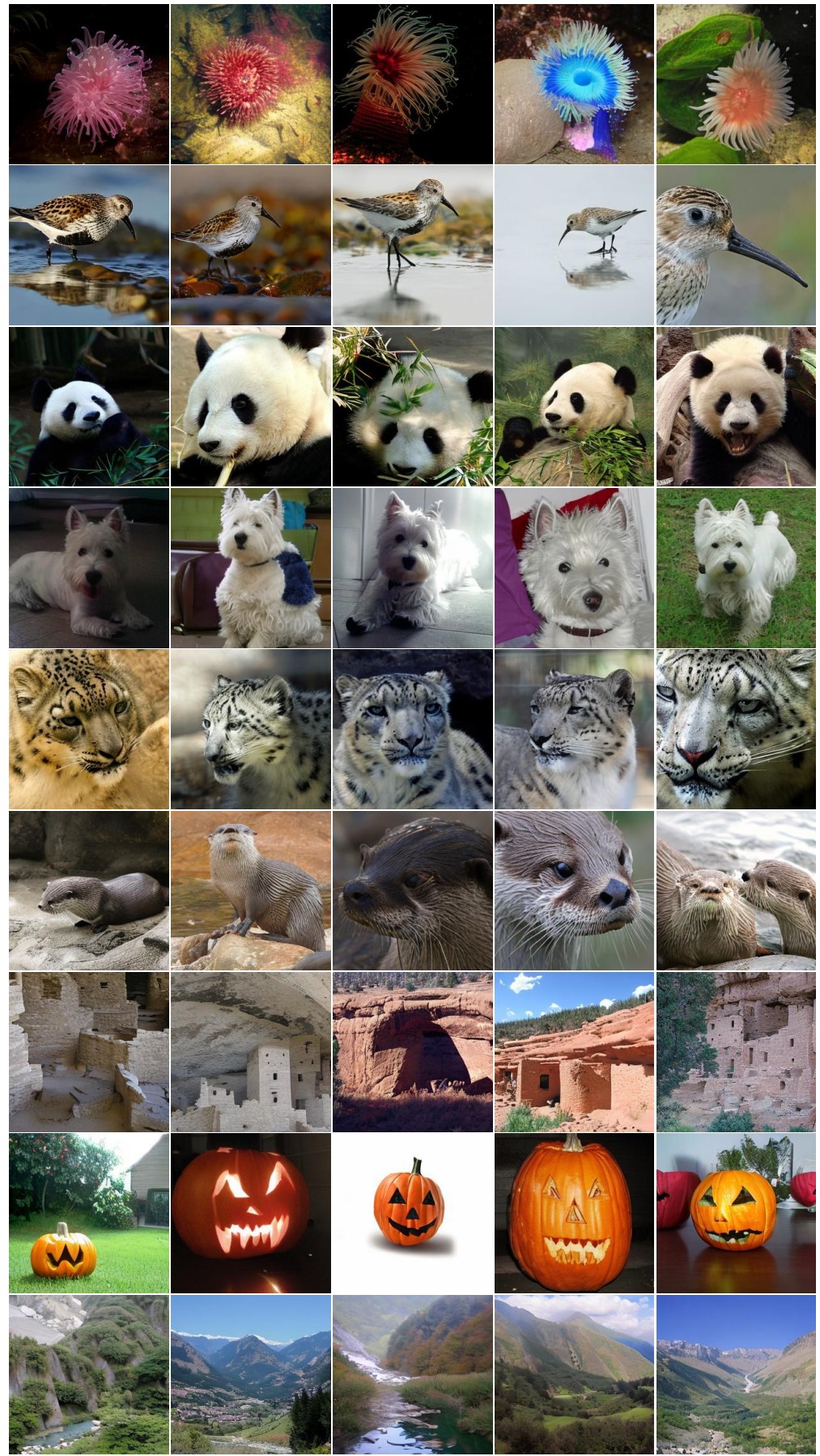

Figure 6: Uncurated samples on ImageNet-256 × 256 with 4-step sampling.

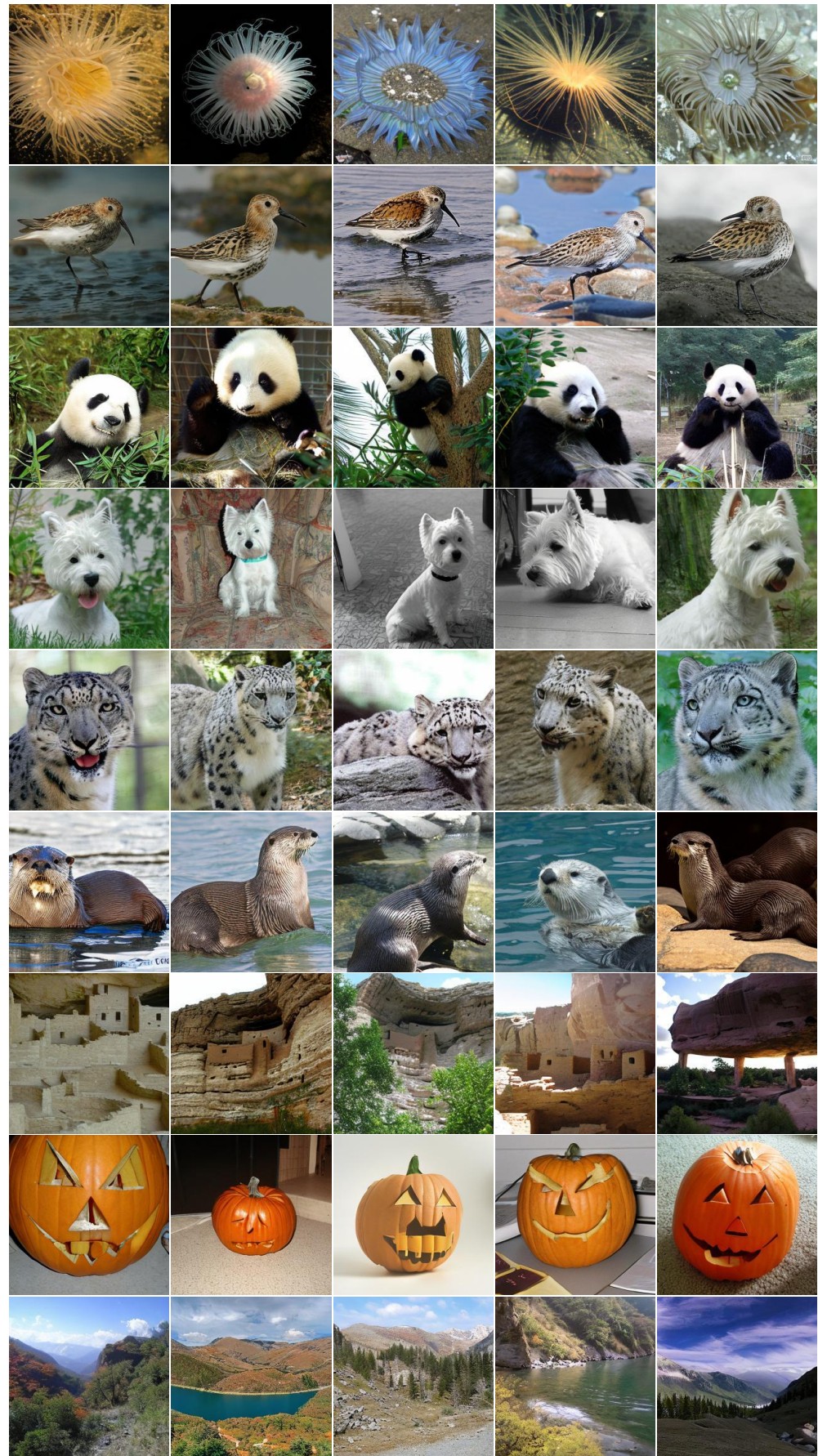

Figure 7: Uncurated samples on ImageNet-256 × 256 with 8-step sampling.

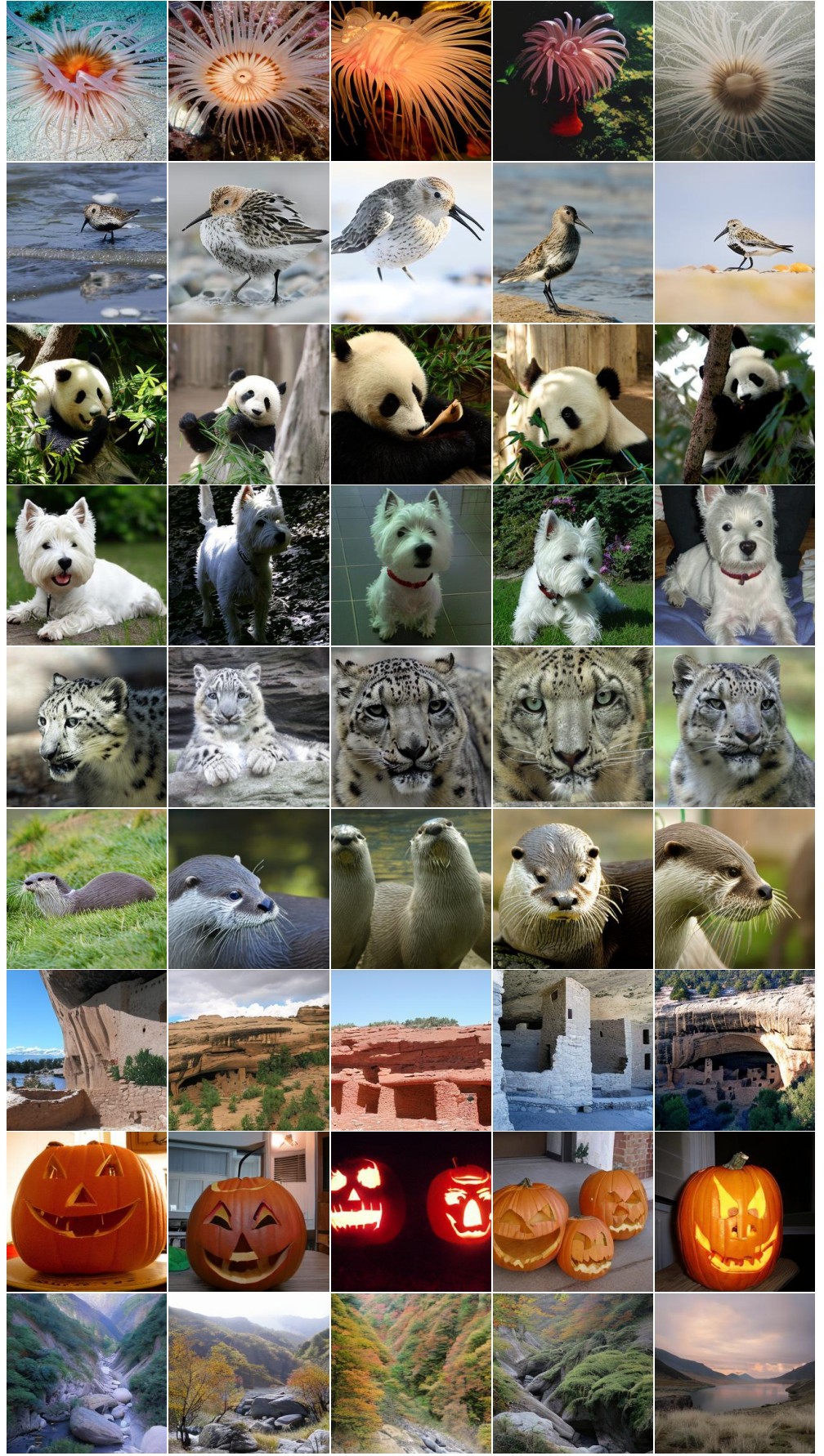

Figure 8: Uncurated samples on ImageNet-256 × 256 with 128-step sampling.

