# OpenReview forum: "Improved Training Technique for Shortcut Models"
_NeurIPS.cc/2025/Conference — NeurIPS 2025 poster_

### Official Review · Reviewer_r2K6 · 2025-06-13

**Clarity:** 3
**Significance:** 4
**Originality:** 4
**Rating:** 5
**Confidence:** 4

**Summary:**

This paper introduces a novel framework called the Improved Shortcut Model (iSM), which aims to overcome the limitations of existing shortcut generative models. While these shortcut models offer efficient few-step generation, they often suffer from issues like fixed guidance strength, high training variance, and a bias towards low-frequency details, which can compromise flexibility, stability, and sample quality. iSM addresses these challenges through several key innovations: it incorporates dynamic guidance by making guidance strength a model input and optimizing its application; it employs a multi-level wavelet loss to better capture crucial high-frequency details; it utilizes scaled optimal transport matching to stabilize training and straighten the generation path; and finally, it implements a dual EMA (Exponential Moving Average) strategy to effectively tackle target lag issues that arise during training.

**Questions:**

- Can the authors provide scaling law experiments for larger models?
- Can the authors add experiments comparing their method with SOTA models across 1 to 50 steps?
- Can the authors provide the complete training resources and more detailed, reproducible experimental steps?

**Ethical Concerns:**

["NO or VERY MINOR ethics concerns only"]

**Final Justification:**

The author has resolved my concerns regarding the model's scalability and reproducibility. I am very interested in the code, which the authors plan to open-source in the future.

**Limitations:**

yes

**Quality:**

3

**Strengths And Weaknesses:**

### **Summary Of Strengths:**

- The paper's core contribution lies in its proposed iSM framework's ability to significantly improve the FID scores of shortcut models when generating images in single-step or few-step (e.g., 4-step) settings. This means it achieves better image quality while maintaining high efficiency, effectively solving the performance degradation issue that shortcut models typically face at low sampling steps.
- By introducing Integrated Guidance and interval guidance, the model allows users to more flexibly adjust guidance strength during inference to balance diversity and fidelity without retraining the model. This overcomes the limitation of previous shortcut models having fixed CFG scales.
- This work doesn't just tackle a single problem; it systematically addresses multiple critical technical challenges that previously hindered shortcut model performance. It achieves this through various innovative methods, including multi-level wavelet loss, scaled optimal transport matching, and a dual EMA strategy, thereby concurrently resolving issues like frequency bias, high training variance, and self-consistency target lag.

### **Summary Of Weaknesses:**

- Experiments were primarily conducted on ImageNet 256x256. While this is a standard benchmark, a more comprehensive evaluation is needed to understand how the method performs on other types of datasets or at higher image resolutions.
- The paper lacks a comparison with SOTA multi-step models. This would be beneficial for researchers to understand the performance gap between few-shot and traditional text-to-image models, such as Stable Diffusion 2.5.
- It appears that the main body of this paper does not include a section on experimental setup, and there is no report on GPU resources in the appendix either. Providing more detailed setup information in the main text would be beneficial for understanding the experiments.

---

> ### Author Rebuttal · Authors · 2025-07-31
>
> Thank you for recognizing the core strengths of our work, including the significant FID improvements, enhanced flexibility, and the systematic approach to solving multiple challenges. We appreciate your suggestions for improvement.
>
> **Note**: References to tables, figures, or propositions without the 'Rebuttal' prefix refer to content located in the main paper.
>
> __Q1: ImageNet 512x512 and different backbone__
>
> Our work introduces a robust, architecture-agnostic training methodology for shortcut models. While we report results on SiT-XL/2 to align with the Shortcut Model baseline, our technique generalizes beyond Transformer-based models. Due to limited time, we tackle both backbone and larger image size within one experiment. Specifically, we **scaled our approach to ImageNet 512×512 using XL/2 variant of FlowDCN**, a fully convolutional generative model with group-wise deformable convolution blocks instead of a DiT-based model. Given the rebuttal time, our models have not been fully trained, and we report their scores at the latest iteration (300K) in the table below. We achieved a substantial FID improvement compared to the original shortcut model training, demonstrating both **scalability and broad applicability**.
>
> **Rebuttal Table 1: ImageNet 512x512 experiment.**
> | Model (FlowDCN) | NFE | #Iter | FID↓  | Precision↑ | Recall↑ |
> | --------------- | --- | ----- | -----| ---------- | ------- |
> | SM              | 1   | 300K  | 43.81  | 0.56       | 0.11    |
> | **iSM (ours)**      | 1   | 300K  | **37.05** | **0.60**        | **0.55**    |
> | SM              | 4   | 300K  | 12.16 | **0.86**       | 0.19    |
> | **iSM (ours)**      | 4   | 300K  | **9.94** | 0.78       | **0.62**    |
>
> __Q2: Text-to-image and compare multi-step model__
>
> We thank the reviewer for pointing out the value of a broader multi-step comparison.  In fact, in Table 1 we compared iSM against the recent multi-step class-conditioned diffusion and flow models on ImageNet 256×256 (e.g. DiT-XL/2 (250 steps), SiT-XL/2 (250 steps), U-ViT-H (50 steps), SimDiff (512 steps),...).
>
> Regarding text-to-image (T2I) models such as Stable Diffusion, we note the following:
>   1.  Our work is scoped to class-conditioned ImageNet generation (as do prior methods like DiT, SiT, SM, IMM).
>   2.  Extending to T2I would require both a text encoder and retraining on large scale dataset like LAION datasets, far beyond our current compute budget.
>   3.  Unlike many text-to-image models, most class-conditional methods do not report, or perform poorly at, the 50-step sampling budget. To our knowledge, U-ViT-H is the only class-to-image model with competitive FID at 50 steps, and we included its result in Table 1 for direct comparison.
>   4.  Architecturally, we build on DiT-based backbones, which have recently been demonstrated (e.g., in PixArt-Alpha) to scale effectively to text conditioning. We therefore believe our improvements would transfer directly to the T2I setting.
>
> In summary, Table 1 shows iSM competitive performance with respect to SOTA class-conditional generative models on ImageNet, and while a direct text-to-image model comparison is an exciting future direction, it lies outside our current class-to-image benchmark and compute resources.
>
> __Q3: Scaling law experiments for iSM__
>
> Due to limited time of rebuttal and computational budget, we couldn't scale up the model to a larger size than DiT-XL, hence we follow SM to run the scaling law experiment from smaller model (DiT-B) to our final model (DiT-XL) as a proxy for larger-scale behavior. The results in Rebuttal Table 2 clearly show that the performance gap between our iSM and the baseline SM widens significantly as model size increases, suggesting strong scaling properties.
>
> **Rebuttal Table 2: Scaling law experiment in ImageNet 256x256.**
> | Backbone   | #Params (M) | NFE | FID (SM) ↓ | FID (iSM) ↓ |
> |------------|-------------|-----|------------|------------ |
> |   DiT-B    | 130         |  1  |    40.31    |   **22.74** |
> |   DiT-B    | 130         |  4  |    28.32    |   **8.69**  |
> |   DiT-XL   | 675         |  1  |    10.60    |   **5.27**  |
> |   DiT-XL   | 675         |  4  |    7.81    |   **2.05**  |
>
>
> __Q4: Provide complete training resources and reproducible steps?__
>
> Yes. We have already included our implementation details in the Appendix and a checklist in the main paper. All experimental workflows, including training and evaluation, were conducted on 8 x NVIDIA A100-40GB GPUs. We are **committed to open-sourcing our code and releasing model checkpoints** upon acceptance to ensure full reproducibility for the community.

---

> ### Comment · Reviewer_r2K6 · 2025-08-07
>
> The author has answered my questions. I believe this method demonstrates good generality and scalability, so I will be raising my score.

---

### Official Review · Reviewer_MiHU · 2025-07-01

**Clarity:** 3
**Significance:** 2
**Originality:** 3
**Rating:** 4
**Confidence:** 4

**Summary:**

This paper proposes several techniques to improve training of shortcut models. Specifically, they introduce 5 improvements: (1) integrate guidance scale as an input to the network, (2) interval guidance, (3) multi-level wavelet loss, (4) twin ema that separates the model weight for inference and self-consistency target, (5) OT matching. By incorporating all of these five components, the authors show shortcut model training can be greatly improved compared with the vanilla model training.

**Questions:**

N/A

**Ethical Concerns:**

["NO or VERY MINOR ethics concerns only"]

**Final Justification:**

I lean toward accepting the paper, though not strongly, as it focuses on technical improvements to shortcut models and limits its scope to class-conditional ImageNet generation. Considering both its empirical and academic contributions, I decide to maintain my current score.

**Limitations:**

Yes

**Paper Formatting Concerns:**

Component-wise analysis seems to follow the format of [1], and I would recommend authors to mention it. Also, it would be better if Table 2 is placed in the upper part of page 9.

[1] Yu et al., Representation Alignment for Generation: Training Diffusion Transformers Is Easier Than You Think

**Quality:**

3

**Strengths And Weaknesses:**

**Strengths**
- The paper is generally well-written and easy to follow.
- Several components looks novel, e.g., applying OT over aggregated multiple minibatches.
- The improvement compared with the vanilla shortcut model seems quite significant.
- The authors provide component-by-component analysis.

**Weaknesses**
- There's no detailed explanation of how sOT is performed. Could the authors provide more detailed explanation (e.g., algorithm)?
- OT will require additional computational and memory burden. The paper mentioned it but it would be great if more explicit analysis (e.g., by numbers) were provided.
- I wonder if Multi-Level Wavelet Function is scalable to resolution, e.g., if it still works with ImageNet 512x512.
- Interval guidance improves the performance, but at the same moment, it also induces a new hyperparameter $t_{\text{interval}$ that should be set at training. While the authors show 0.3 is optimal in their experiments, but experiments on other setups (e.g., other datasets or higher resolution datasets) would be make the paper more stronger by showing such values are robust to different setups.
- The paper only uses FID for quantitative evaluation. Other metrics (e.g., IS, Precision, Recall) can be provided.

---

> ### Author Rebuttal · Authors · 2025-07-31
>
> Thank you for the positive and constructive review. We appreciate that you found the paper well-written, the components novel, and the improvements significant. We address your points below.
>
> **Note**: References to tables, figures, or propositions without the 'Rebuttal' prefix refer to content located in the main paper.
>
> __Q1: Experiments on ImageNet 512x512, other backbones, and generalizability.__
>
> Our work introduces a robust, architecture-agnostic training methodology for shortcut models. While we report results on SiT-XL/2 to align with the Shortcut Model baseline, our technique generalizes beyond Transformer-based models. Due to limited time, we tackle both backbone and larger image size within one experiment. Specifically, we **scaled our approach to ImageNet 512×512 using XL/2 variant of FlowDCN**, a fully convolutional generative model with group-wise deformable convolution blocks instead of a DiT-based model. Given the rebuttal time, our models have not been fully trained, and we report their scores at the latest iteration (300K) in the table below. We achieved a substantial FID improvement compared to the original shortcut model training, demonstrating both **scalability and broad applicability**.
>
> **Rebuttal Table 1: ImageNet 512x512 experiment.**
> |Model(FlowDCN)|NFE|#Iter|FID↓|Precision↑|Recall↑|
> |---|---|---|---|---|---|
> |SM|1|300K|43.81|0.56|0.11|
> |**iSM(ours)**|1|300K|**37.05**|**0.60**|**0.55**|
> |SM|4|300K|12.16|**0.86**|0.19|
> |**iSM(ours)**|4|300K|**9.94**|0.78|**0.62**|
>
> In line with common practice in diffusion model research, our work focuses on class-to-image tasks, as scaling to full text-to-image generation is challenging under current resource and time constraints. Many influential papers in this field, such as DiT, SiT, EDM, Consistency Models, and Shortcut Models, have **established their contributions through experiments on label-to-image generation**, thereby laying the groundwork for future scaling to text-to-image diffusion models.
>
> __Q2: Detailed algorithm for sOT__
>
> The procedure is as follows:
> 1.  **Aggregate Batches:** For every `K` training steps, we collect and hold the noise samples and data latents from `K` separate mini-batches of size `M`.
> 2.  **Compute Large-Scale OT Plan:** We compute a single OT plan over the aggregated set of `K*M` samples.
> 3.  **Redistribute and Train:** The resulting `K*M` optimally paired samples are then split back into `K` mini-batches, which are used for the subsequent `K` model training steps.
>
> This process allows the model to benefit from a large-scale transport plan without the memory burden of a large training batch size. We will add pseudo-code algorithm for Scaling Optimal Transport (sOT) into the paper.
>
> __Q3: Computational and memory overhead of sOT__
>
> The computational overhead from sOT is a modest **~4%** increase in total training time, while the memory overhead is negligible.
>
> - **Memory Efficiency:** Our sOT implementation is memory-efficient because it **decouples the OT batch size (`K*M`) from the model's training batch size (`M`)**. We only need to store the `K*M` latent vectors, while the memory-intensive model weights and gradients are processed with the standard, smaller batch size `M`.
> - **Computational Cost:** For `K=32` and a per-GPU batch size of `M=64`, the OT plan is computed over 2048 samples every 32 steps. This adds only ~4% to the total training time on an A100 GPU, a small price for the significant gains in trajectory stability and final FID scores. We report the training iteration speed for different values of K (where K=0 indicates no OT used).
>
> **Rebuttal Table 2: Training speed with sOT in ImageNet 256x256.**
> | K | 0 | 1 | 8 | 32 |
> | :- | :- | :- | :- | :- |
> | **Training speed (s/iter)** | 0.361 | 0.362 | 0.368 | 0.376 |
>
> **Q4: I wonder if Multi-Level Wavelet Function is scalable to resolution, e.g., if it still works with ImageNet 512x512.**
>
> Yes, the Multi-Level Wavelet Loss is highly scalable and particularly well-suited for higher resolutions. The Discrete Wavelet Transform (DWT) is an inherently multi-scale algorithm. Its ability to disentangle features across frequency bands becomes **even more critical** for high-resolution images, where preserving fine-grained textures (high-frequency) alongside global composition (low-frequency) is essential for quality. This makes it a robust choice ([1], [2]) for scaling to 512x512 and beyond. Our result in Rebuttal Table 1 on ImageNet 512x512 also strengthen this argument, since iSM significantly outperforms the original shortcut model.
>
> __Q5: Robustness of the interval guidance hyperparameter__
>
> We affirm that $t_{\text{interval}}$ is a robust hyperparameter, a finding supported by theoretical motivation, our internal ablation studies, and external validation from concurrent work.
>
> 1.  **SNR-Based Principle:** The parameter is not arbitrary but is grounded in the signal-to-noise ratio (SNR) dynamics of diffusion models. Notably, previous work [1] found that guidance is ineffective or harmful at very high noise levels (low $t$), and $t_{\text{interval}}$ simply implements this principle.
>
> 2.  **Empirical Robustness (Table 2):** Our ablation study demonstrates a wide effective range, not a single brittle value.
> - Even a small cutoff at $t_{\text{interval}} = 0.1$ substantially improves performance ($\text{FID}_{N=1}$ drops from 9.62 to 8.58), showing the primary benefit is avoiding guidance in the earliest, highest-noise steps.
> - Performance degrades only when the cutoff is overly aggressive (e.g., $t_{\text{interval}} = 0.5$), confirming a **stable optimal window** rather than a specific magic number.
> - Furthermore, we have trained a new model (different architecture, FlowDCN) and new dataset (ImageNet 512x512) with the same $t_{\text{interval}}=0.3$ and the performance is still very competitive (in Rebuttal Table 1)
>
> 3.  **Consistency with Prior Work:** Our finding is strongly validated by prior work [1], show this principle is critical for state-of-the-art results on diverse setups like ImageNet-512 and Stable Diffusion XL. Both our work (integrating interval guidance into the training loss) and theirs (applying it at inference) identify a similar optimal range around $t=0.3$.
>
> This convergence of evidence confirms that $t_{\text{interval}}=0.3$ is a robust and principled default, not a dataset-specific choice.
>
> __Q6: Additional evaluation metrics (FD-DINOv2, IS, Precision, Recall)__
>
> We report FD-DINOv2, Inception Score (IS), Precision, and Recall for the baseline Shortcut Model (SM), Inductive Moment Matching (IMM), and our improved Shortcut Model (iSM). The results below show that **iSM consistently outperforms the baseline SM** across multiple NFE settings.
>
> **Rebuttal Table 3: Additional evaluation metrics for ImageNet 256x256.**
> | Model | NFE | FD-DINOv2↓ | IS↑ | Precision↑ | Recall↑ |
> | :--- | :--- | :--- | :--- | :--- | :--- |
> | SM | 1 |  500.92 | 102.66 | 0.68 | 0.46 |
> | IMM | 1 |  247.78 | 128.87 | 0.67 | **0.64** |
> | **iSM (ours)** | 1 |  **232.31** | **223.52** | 0.70 | 0.61 |
> | SM | 2 |  329.53 | 125.66 | **0.77** | 0.47 |
> | IMM | 2 | 152.08 | 173.66 | 0.74 | **0.64** |
> | **iSM (ours)** | 2 | **107.63** | **302.29** | **0.77** | 0.61 |
> | SM | 4 | 265.90 | 136.79 | **0.80** | 0.48 |
> | IMM | 4 | 110.88 | 204.95 | 0.77 | 0.62 |
> | **iSM (ours)** | 4 | **83.70** | **298.23** | 0.77 | **0.62** |
>
> __Q7: Regarding Formatting Concerns:__
>
> Thank you for these helpful suggestions. We will them in the revision:
> *   We will cite **REPA [3]** in our component-wise analysis section as an inspiration for the evaluation structure.
> *   We will reposition **Table 2** to the top of Page 9 for improved readability and flow.
>
> [1] Applying Guidance in a Limited Interval Improves
> Sample and Distribution Quality in Diffusion Models, NeurIPS 2024.
>
> [2] Diffusion-4K: Ultra-High-Resolution Image Synthesis with Latent Diffusion Models, CVPR 2025.
>
> [3] Representation alignment for generation: Training diffusion transformers is easier than you think, ICLR 2025.

---

> > ### Comment · Reviewer_MiHU · 2025-08-03
> > **Response**
> >
> > Thanks for your detailed response. Many of my concerns have been addressed. Regarding text-to-image generation, I understand that prior one-step generation models have not considered it. Nevertheless, I strongly encourage the authors to explore and include it in the final manuscript (not necessarily during the rebuttal period), as it would offer valuable insights into the scalability of shortcut models in the widely used conditional image generation setting. Performing T2I experiments on MS-COCO is relatively inexpensive (even less demanding than ImageNet-256), and the authors can follow the setup in U-ViT or REPA.

---

### Official Review · Reviewer_DUtt · 2025-07-02

**Clarity:** 3
**Significance:** 3
**Originality:** 2
**Rating:** 5
**Confidence:** 4

**Summary:**

The authors introduce multiple numerical tricks to make class-conditional consistency models competitive with standard class-conditional generative models. The tricks are the following: a wavelet-based loss, a ¨scaled¨ optimal transport flow matching, and leveraging EMA networks with multiple decays (a slow decay for inference, and a large decay to anchor the consistency model).

**Questions:**

- ¨A critical, previously overlooked issue in shortcut models is the
accumulation of guidance effects.¨ Is this a well known problem? If yes can authors provide a reference on it? if not, is it a contribution?


Ablation study.
I love the fact authors provide a detailed ablation study, however I have some questions about it:
- Some of the performance gain claims seems rather small and might not be significative, especially for the ¨augmented OT¨ batch parameter K. To the best of my knowledge, OT flow matching makes bring empirical improvement when one can only use a small number of inference steps (NFE<10, which makes sense in your setting), and does not bring improvement with a large number of inference steps (NFE> 50). However, computing OT pairing in the 256*256 image space should not make sens: the dimension is too large compared to the number of samples to give meaningful information. I wonder if authors have any kind of intuition on this?
- If understand well your benchmark, GANs is still the best 1-step generation model for conditional generation?



- Are Section 4-6 contribution of the paper?


Multi-level Wavelet Function:
- Could authors provide additional information on the ¨multi-level Wavelet Function¨? Is it mainly a modification of the loss?
- Do you use a latent space? as in [3]?


[1] Stein, George, et al. "Exposing flaws of generative model evaluation metrics and their unfair treatment of diffusion models." Advances in Neural Information Processing Systems

[2] Sajjadi, Mehdi SM, et al. "Assessing generative models via precision and recall." Advances in neural information processing systems 31 (2018).

[3] Rombach, Robin, et al. "High-resolution image synthesis with latent diffusion models." Proceedings of the IEEE/CVF conference on computer vision and pattern recognition

**Ethical Concerns:**

["NO or VERY MINOR ethics concerns only"]

**Final Justification:**

The authors answered my concerns and thus I raised my score.

**Limitations:**

yes

**Quality:**

3

**Strengths And Weaknesses:**

Strengths:
- The paper is easy to follow and tackles an important problem

Weakness:
- Given the performance improvements (all methods perform very similarly), I would recommend to add other performance metrics, such as the FID with DINOv2 embedding (as recommended in [1], instead of the inception network), maybe also the precision and recall metrics [2]?

---

> ### Author Rebuttal · Authors · 2025-07-31
>
> Thank you for your valuable feedback and questions. We are glad you found the paper easy to follow and the problem important.
>
> **Note**: References to tables, figures, or propositions without the 'Rebuttal' prefix refer to content located in the main paper.
>
> __Q1: Additional evaluation metrics (FD-DINOv2, IS, Precision, Recall)__
>
> We report FD-DINOv2[1], Inception Score[2] (IS), Precision[3], and Recall[3] for the baseline Shortcut Model (SM), Inductive Moment Matching (IMM), and our improved Shortcut Model (iSM). The results below show that **iSM consistently outperforms the baseline SM** across multiple NFE settings.
>
> **Rebuttal Table 1: Additional evaluation metrics for ImageNet 256x256.**
> | Model | NFE | FD-DINOv2↓ | IS↑ | Precision↑ | Recall↑ |
> | :--- | :--- | :--- | :--- | :--- | :--- |
> | StyleGAN-XL | 1 |  **210.49** | **264.86** | **0.80** | 0.50 |
> | SM | 1 |  500.92 | 102.66 | 0.68 | 0.46 |
> | IMM | 1 |  247.78 | 128.87 | 0.67 | **0.64** |
> | **iSM (ours)** | 1 |  232.31 | 223.52 | 0.70 | 0.61 |
> | SM | 2 |  329.53 | 125.66 | **0.77** | 0.47 |
> | IMM | 2 | 152.08 | 173.66 | 0.74 | **0.64** |
> | **iSM (ours)** | 2 | **107.63** | **302.29** | **0.77** | 0.61 |
> | SM | 4 | 265.90 | 136.79 | **0.80** | 0.48 |
> | IMM | 4 | 110.88 | 204.95 | 0.77 | 0.62 |
> | **iSM (ours)** | 4 | **83.70** | **298.23** | 0.77 | **0.62** |
>
>
> __Q2: Is the "accumulated guidance" issue a well-known problem? Is it a contribution?__
>
> To the best of our knowledge, the **explicit formalization and analysis** of the accumulation of guidance effects in shortcut models is a **novel contribution** of our paper. While CFG is well-studied, its compounding effect in shortcut models has been overlooked. We are the **first to identify, formalize (Proposition 1), and solve this critical flaw**. We’ll revise the introduction to emphasize this contribution, making its importance immediately clear.
>
> __Q3: Significance of OT gains and intuition for high-dimensional OT__
>
> 1.  **OT operates in Latent Space:** We agree that OT in 256x256 pixel space may be intractable. **In fact, all our operations, including OT matching, are performed in the 32x32 latent space** of a pre-trained VAE. We apologize if this was not clear.
> 2.  **Significance of Gains:** In the high-performance regime (FID < 3.0), improvements are non-linear and dropping from **2.64 to 2.23 is a major leap**, since each FID reduction in this regime becomes increasingly challenging (for example, in the paper SiT's Table 1, last two rows, when applying CFG, the improvement drop from 2.27 to 2.06)
>
> __Q4: If I understand well your benchmark, GANs is still the best 1-step generation model for conditional generation?__
>
> It’s true that state-of-the-art GANs achieve the lowest FID in a single step, but they’re notoriously hard to train and often suffer from **diversity collapse**. In contrast, our diffusion-based iSM **trains stably** (no adversarial loss) and delivers **significantly better sample diversity** (Recall of 0.61 vs. StyleGAN-XL of 0.50), narrowing the FID gap while ensuring a **stronger diversity–fidelity trade-off**. Furthermore, iSM’s **flexible sampling schedules** and efficient guidance sampling set it apart from other diffusion models.
>
> __Q5: Are Sections 4-6 a contribution of the paper?__
>
> Absolutely. In fact, in the Introduction (Sec. 1, lines 53–76) we explicitly identify these shortcomings of vanilla shortcut models: (1) low-frequency bias, (2) high-variance/curved trajectories, and (3) EMA-lagged self-consistency, and then state our three complementary contributions to address them:
> - Multi-Level Wavelet Loss (Section 4) to correct frequency bias;
> - Scaling Optimal Transport Matching (Section 5) to straighten and stabilize trajectories;
> - Twin EMA Strategy (Section 6) to remove the temporal lag in self-consistency targets.
>
> These sections detail each of our contributions. As shown in Table 2, progressively adding the components from Sec. 4–6 steadily improves the model’s performance, confirming their novelty and importance.
>
> __Q6: Could authors provide additional information on the "Multi-Level Wavelet Function"?__
>
> Certainly. The Multi-Level Wavelet Loss functions as a **multi-scale objective** that tackles the frequency bias. Instead of naively applying L2 loss in the spatial domain, it **recursively** applies Discrete Wavelet Transform (DWT) to the prediction and target. This decomposes the error signal across various frequency bands, forcing the network to minimize errors in both low-frequency structures and high-frequency details simultaneously. For reference, Appendix B provides the pseudocode (Algorithm 1).
>
> __Q7: Do you use a latent space?__
>
> Yes. Following previous works like DiT, SiT, SM, our model operates entirely in the latent space of the pre-trained `sd-vae-ft-mse` autoencoder. We will make this detail more prominent in the main text to avoid any confusion.
>
> [1] Exposing flaws of generative model evaluation metrics and their unfair treatment of diffusion models, NeurIPS 2023.
>
> [2] Improved techniques for training gans, NeurIPS 2016.
>
> [3] Improved Precision and Recall Metric for Assessing Generative Models, NeurIPS 2019.

---

> > ### Comment · Reviewer_DUtt · 2025-08-01
> >
> > The authors addressed my concerns, and as a result, I raised my score.

---

### Official Review · Reviewer_xmQN · 2025-07-03

**Clarity:** 3
**Significance:** 3
**Originality:** 3
**Rating:** 5
**Confidence:** 4

**Summary:**

This paper presents an Improved Shortcut Model (iSM) to address key limitations of existing shortcut models in generative modeling. The identified issues include fixed classifier-free guidance (CFG) scales limiting inference flexibility, high training variance from random noise-data pairings, frequency bias toward low-frequency components, and conflicts between EMA updates and self-consistency objectives. To tackle these, the authors propose a unified framework incorporating: (1) Integrated Guidance for dynamic guidance scaling at inference; (2) Multi-Level Wavelet Loss to mitigate frequency bias; (3) Scaling Optimal Transport (sOT) Matching to reduce training variance; and (4) a Twin EMA strategy to balance self-consistency and training stability. Extensive experiments on ImageNet 256×256 demonstrate that iSM achieves significant FID improvements across on1e-step, few-step, and multi-step generation, outperforming baseline shortcut models and comparable methods.

**Questions:**

1. How about other metrics for comparison, such as LPIPS or IS?
2. The Scaling OT (sOT) method is very interesting. Have you investigated the effect of the update frequency? You currently recompute it every K batches; would performance change if it were recomputed more or less frequently, and how does this interact with the scaling factor K?
3. The accumulated guidance problem is formalized for N=128 base steps. Does this problem persist for other base step sizes (e.g., N=64 or N=256)? How does iSM’s Integrated Guidance adapt to varying N?
4. Regarding the Twin EMA strategy, the choice of a fast decay rate of 0.95 for the target network seems relatively low compared to the slow rate of 0.999. Could you provide more intuition on how you arrived at this value? Is there a risk of the target network becoming too noisy or unstable with such a fast update?

**Ethical Concerns:**

["NO or VERY MINOR ethics concerns only"]

**Final Justification:**

The author's response has addressed all my concerns. After carefully reading the comments from other reviewers, I believe this work deserves acceptance, and therefore, I have decided to further raise my rating to a clear accept.

**Limitations:**

1. The gap between one-step and multi-step is still significant, which is against the claim of this paper.
2. Lack large-scale experiments to further support the claims of this paper.

**Quality:**

3

**Strengths And Weaknesses:**

Strengths:
1. The paper identifies both known (e.g., fixed CFG scales) and overlooked (e.g., accumulated guidance effects, EMA-self-consistency conflict) limitations of shortcut models, providing a clear motivation for the proposed improvements.
2. The proposed four core components in this paper (Integrated Guidance, Multi-Level Wavelet Loss, sOT, Twin EMA) are complementary, forming a cohesive solution that addresses multiple issues simultnaeously.
3. The authors conduct thorough experiments, including quantitative comparisons with state-of-the-art models, ablation studies for each component, and analysis of key hyper-parameters.
4. The proposed iSM supports dynamic guidance scaling and reduces inference time by ~50% compared to standard CFG, improving real-world applicability for resource-constrained scenarios.


Weaknesses:
1. All experiments are conducted on ImageNet 256×256 and this paper lacks further solid experimental evaluations on larger-scale experiments, such as ImageNet 512x512, or other text-to-image experiments. It remains unknown how well these techniques and hyper-parameters perform on more complex generation tasks.
2. Though the proposed method iSM reduces inference steps, the performance is still limited compared to other cutting-edge models, such as REPA-XL/2 (FID=1.42) (compared to iSM-128NFE, 1.88 FID). In addition, it still lags behind multi-step methods (FID 5.27 vs. 1.88 in 128 steps). The gap is significant.
3. The proposed iSM is tested only with the SiT-XL/2 architecture; its compatibility with other backbones (e.g., DiT, U-ViT) is unproven, limiting its generalizability.

---

> ### Author Rebuttal · Authors · 2025-07-31
>
> Thank you for your detailed review and constructive feedback. We appreciate that you found our motivation clear, our proposed solution cohesive, and our experiments thorough. Below, we provide a point-to-point response and a summary of the corresponding revisions.
>
> **Note**: References to tables, figures, or propositions without the 'Rebuttal' prefix refer to content located in the main paper.
>
> **Q1: Experiments on ImageNet 512x512, other backbones, and generalizability.**
>
> Our work introduces a robust, architecture-agnostic training methodology for shortcut models. While we report results on SiT-XL/2 to align with the Shortcut Model baseline, our technique generalizes beyond Transformer-based models. Due to limited time, we tackle both backbone and larger image size within one experiment. Specifically, we **scaled our approach to ImageNet 512×512 using XL/2 variant of FlowDCN**, a fully convolutional generative model with group-wise deformable convolution blocks instead of a DiT-based model. Given the rebuttal time, our models have not been fully trained, and we report their scores at the latest iteration (300K) in the table below. We achieved a substantial FID improvement compared to the original shortcut model training, demonstrating both **scalability and broad applicability**.
>
> **Rebuttal Table 1: ImageNet 512x512 experiment.**
> |Model(FlowDCN)|NFE|#Iter|FID↓|Precision↑|Recall↑|
> |---|---|---|---|---|---|
> |SM|1|300K|43.81|0.56|0.11|
> |**iSM(ours)**|1|300K|**37.05**|**0.60**|**0.55**|
> |SM|4|300K|12.16|**0.86**|0.19|
> |**iSM(ours)**|4|300K|**9.94**|0.78|**0.62**|
>
> In line with common practice in diffusion model research, our work focuses on class-to-image tasks, as scaling to full text-to-image generation is challenging under current resource and time constraints. Many influential papers in this field, such as DiT, SiT, EDM, Consistency Models, and Shortcut Models, have **established their contributions through experiments on label-to-image generation**, thereby laying the groundwork for future scaling to text-to-image diffusion models.
>
> __Q2: Comparison with other cutting-edge models__
>
> A direct FID comparison between iSM and models like REPA can be misleading, as they operate in fundamentally different **inference-efficiency regimes**. REPA is a state-of-the-art **full-trajectory** diffusion model requiring 250 NFE to achieve its FID of 1.42. Our iSM achieves a competitive FID of **1.93 with just 8 NFE**. This represents a **>30x reduction in computational cost** for the trade-off in final FID. Our contribution lies in **improving shortcut models across at extreme NFE efficiency**, achieving a significant FID reduction in these settings over the original baseline and competitive performance with other model families.
>
> __Q3: Additional evaluation metrics (FD-DINOv2, IS, Precision, Recall)__
>
> We report FD-DINOv2, Inception Score (IS), Precision, Recall for the baseline Shortcut Model (SM), Inductive Moment Matching (IMM), and our improved Shortcut Model (iSM). The results below show that iSM consistently outperforms the baseline SM across multiple NFE settings.
>
> **Rebuttal Table 2: Additional metrics for ImageNet 256x256.**
> |Model|NFE|FD-DINOv2↓|IS↑|Precision↑|Recall↑|
> |:---|:---|:---|:---|:---|:---|
> |SM|1|500.92|102.66|0.68|0.46|
> |IMM|1|247.78|128.87|0.67|**0.64**|
> |**iSM (ours)**|1|**232.31**|**223.52**|0.70|0.61|
> |SM|2|329.53|125.66|**0.77**|0.47|
> |IMM|2|152.08|173.66|0.74|**0.64**|
> |**iSM (ours)**|2|**107.63**|**302.29**|**0.77**|0.61|
> |SM|4|265.90|136.79|**0.80**|0.48|
> |IMM|4|110.88|204.95|0.77|0.62|
> |**iSM (ours)**|4|**83.70**|**298.23**|0.77|**0.62**|
>
>
> __Q4: The Scaling OT (sOT) method is very interesting. Have you investigated the effect of the update frequency?__
>
> Yes, we performed an ablation over the update interval K to measure its impact (see Table 2 in main paper). We found:
> - Better training: Increasing K yields a closer match to the global transport plan, which straightens training trajectories and boosts stability.
> - Minimal overhead: Even at a larger K, the extra overhead remains modest (~4% in our final setup, K = 32).
> - Saturation point: As shown in Table 2, the FID scores improvements plateau past a certain K, so there’s little benefit in updating more frequently.
>
> __Q5: Does the accumulated guidance problem persist for other base step sizes (N)?__
>
> Yes, and we emphasize that this is one of our key contributions. The accumulated guidance problem is a **systemic flaw** in the recursive formulation of shortcut models. Our **Proposition 1** provides the first formalization of this compounding error, showing that the effective guidance scales undesirably as $w^{log_2(N)}$.
>
> Our **Integrated Guidance** framework resolves this issue by making the guidance as an explicit model input `w`, we train the network to learn the mapping `(x, t, c, d, w) -> v_guided` directly. This **bypasses the flawed recursive formulation entirely**, making our solution inherently robust and independent of the base step size `N`. We’ll underscore this contribution in Section 3.2.
>
>
> __Q6: Intuition for Twin EMA decay rate (0.95)__
>
> The selection for decay rate for fast-decay EMA ($\theta_{target}$) needs to satisfy the following:
> - **Low enough** to keep the self-consistency target network closely aligned with the online network, avoiding the distributional lag as training progresses.
> - **High enough** to preserve the benefits of EMA, which smooths noisy mini-batch gradients and prevents training instability.
>
>
> Therefore, a much lower rate decay for $\theta_{target}$ risks instabilility and negates the benefits of EMA, while a much higher rate negates the benefit of the twin strategy and reintroduces distributional lags. Our ablation in Table 2 empirically validates that 0.95 provides the optimal trade-off. The decay rate of 0.95 allows the self-consistency target to closely track the online network, but with just enough lag to smooth out high-frequency noise in the training signal.
>
>
> __Q7: The gap between one-step and multi-step performance__
>
> We would like to clarify that our paper does not claim to completely eliminate the gap between one-step and multi-step generation. Instead, our central contribution is to **significantly reduce the performance degradation at lower NFEs** compared to previous shortcut models, while also providing a single flexible model that performs well across a wide range of sampling settings. We identified the degradation as primarily resulting from: (1) accumulated guidance, (2) low-frequency bias, (3) high-variance/curved trajectories, and (4) EMA-lagged self-consistency. For instance, our iSM model improves the FID from a much higher value in SM (10.60) down to **5.27 for one-step** and further down to **2.05 for four-step generation**. As noted in our conclusion (L381-383), while bridging this gap remains a challenging task, it offers an exciting and promising avenue for future research.

---

> > ### Author Response · Authors · 2025-08-09
> > **Gentle Reminder**
> >
> > Dear Reviewer xmQN,
> >
> > Thank you again for your time and efforts in reviewing our work.
> > This is a kind reminder that the discussion phase will be ending soon. We would be happy to address any further questions you may have before it closes.
> >
> > Kind regards,
> >
> > Authors

---

### Decision · Program_Chairs · 2025-09-17

**Decision:**

Accept (poster)

**Comment:**

This paper analyzes and addresses several shortcomings of shortcut models, generative models that allow one- or few-step sampling.

Strengths:

- The paper addresses an important problem (reviewers DUtt, MiHU, r2K6).
- The paper analyses and proposes/formalizes novel solutions to several issues with shortcut models (reviewers xmQN, MiHU, r2K6).
- The experiments confirm the benefits of the proposed approach.

Weaknesses:

- At submission time, the paper only included ImageNet 256x256 (reviewers xmQN, MiHU, r2K6). Authors added ImageNet 512x512 and additional metrics to existing experiments in the rebuttal.
- This is not the end of the story: More steps are still better (reviewer r2K6), and so are GANs (reviewers DUtt). So it's not the end of the story.

All reviewers agreed that the methods and experimental evidence is significant to warrant acceptance. The authors have committed to open-sourcing code and model checkpoints.